# Binding affinity landscapes constrain the evolution of broadly neutralizing anti-influenza antibodies

Angela M Phillips[1†], Katherine R Lawrence[1,2,3,4†], Alief Moulana[1], Thomas Dupic[1], Jeffrey Chang[5], Milo S Johnson[1], Ivana Cvijovic[6], Thierry Mora[7], Aleksandra M Walczak[7], Michael M Desai[1,2,3,5]*

[1]Department of Organismic and Evolutionary Biology, Harvard University, Cambridge, United States; [2]NSF-Simons Center for Mathematical and Statistical Analysis of Biology, Harvard University, Cambridge, United States; [3]Quantitative Biology Initiative, Harvard University, Cambridge, United States; [4]Department of Physics, Massachusetts Institute of Technology, Cambridge, United States; [5]Department of Physics, Harvard University, Cambridge, United States; [6]Department of Applied Physics, Stanford University, Stanford, United States; [7]Laboratoire de physique de l'École Normale Supérieure, CNRS, PSL University, Sorbonne Université, and Université de Paris, Paris, France

*For correspondence:
mdesai@oeb.harvard.edu

†These authors contributed equally to this work

**Abstract** Over the past two decades, several broadly neutralizing antibodies (bnAbs) that confer protection against diverse influenza strains have been isolated. Structural and biochemical characterization of these bnAbs has provided molecular insight into how they bind distinct antigens. However, our understanding of the evolutionary pathways leading to bnAbs, and thus how best to elicit them, remains limited. Here, we measure equilibrium dissociation constants of combinatorially complete mutational libraries for two naturally isolated influenza bnAbs (CR9114, 16 heavy-chain mutations; CR6261, 11 heavy-chain mutations), reconstructing all possible evolutionary intermediates back to the unmutated germline sequences. We find that these two libraries exhibit strikingly different patterns of breadth: while many variants of CR6261 display moderate affinity to diverse antigens, those of CR9114 display appreciable affinity only in specific, nested combinations. By examining the extensive pairwise and higher order epistasis between mutations, we find key sites with strong synergistic interactions that are highly similar across antigens for CR6261 and different for CR9114. Together, these features of the binding affinity landscapes strongly favor sequential acquisition of affinity to diverse antigens for CR9114, while the acquisition of breadth to more similar antigens for CR6261 is less constrained. These results, if generalizable to other bnAbs, may explain the molecular basis for the widespread observation that sequential exposure favors greater breadth, and such mechanistic insight will be essential for predicting and eliciting broadly protective immune responses.

## Introduction

Vaccination harnesses the adaptive immune system, which responds to new pathogens by mutating antibody-encoding genes and selecting for variants that bind the pathogen of interest. However, influenza remains a challenging target for immunization: most antibodies elicited by vaccines provide protection against only a subset of strains, largely due to the rapid evolution of the influenza surface protein hemagglutinin (HA) (*Wiley et al., 1981*; *Smith et al., 2004*). After nearly two decades of studies, numerous broadly neutralizing antibodies (bnAbs) have been isolated from humans, with

varying degrees of cross-protection against diverse strains (*Corti et al., 2017*; *Throsby et al., 2008*; *Dreyfus et al., 2012*; *Corti et al., 2011*; *Schmidt et al., 2015*). Still, we do not fully understand many factors affecting how and when bnAbs are produced. In particular, affinity is acquired through a complex process of mutation and selection (*Victora and Nussenzweig, 2012*), but the effects of mutations on binding affinity to diverse antigens are not well characterized.

For example, consider two well-studied influenza bnAbs that display varying levels of breadth: CR9114 is one of the broadest anti-influenza antibodies ever found, neutralizing strains from both groups of influenza A and strains from influenza B, while CR6261 is limited to neutralizing strains from Group 1 of influenza A (*Throsby et al., 2008*; *Dreyfus et al., 2012*; *Ekiert et al., 2009*; *Lingwood et al., 2012*). Both antibodies were isolated from vaccinated donors, derive from very similar germline sequences (IGHV1 –69 and IGHJ6), and bind the conserved HA stem epitope (*Figure 1—figure supplement 3*; *Throsby et al., 2008*; *Dreyfus et al., 2012*; *Ekiert et al., 2009*). Each antibody heavy chain has many mutations (18 amino acid changes for CR9114, 14 for CR6261, *Figure 1A*), including seven positions that are mutated in both, yet the contributions of these mutations to affinity against different antigens remain unclear (*Dreyfus et al., 2012*; *Avnir et al., 2014*).

Beyond single mutational effects, it remains unknown whether there are correlated effects or strong trade-offs between binding to different antigens (pleiotropy), or non-additive interactions between mutations (epistasis). Such epistatic and pleiotropic effects can constrain the mutational pathways accessible under selection, as has been observed for other proteins (*Weinreich et al., 2006*; *Starr et al., 2017*; *Ortlund et al., 2007*; *Podgornaia and Laub, 2015*; *Gong et al., 2013*; *Sailer and Harms, 2017b*; *Miton and Tokuriki, 2016*; *Poelwijk et al., 2019*; *Bank et al., 2015*). Epistasis in antibody-antigen interactions remains significantly understudied (*Adams et al., 2019*; *Pappas et al., 2014*; *Braden et al., 1998*) and most deep mutational scanning studies have focused on antigens (*Doud et al., 2018*; *Wu et al., 2020*; *Starr et al., 2021*). In contrast to typical protein evolution, antibody affinity maturation proceeds by discrete rounds of mutation and selection (*Victora and Nussenzweig, 2012*), typically with more than one nucleotide mutation occurring between selective rounds (*Unniraman and Schatz, 2007*). In addition, antibodies are inherently mutationally tolerant (*Braden et al., 1998*; *Chen et al., 1999*; *Burks et al., 1997*; *Corti and Lanzavecchia, 2013*; *Klein et al., 2013*), generating opportunities for interactions that scale combinatorially. Thus, if epistatic and pleiotropic constraints exist for antibodies, they could affect the likelihood of producing bnAbs under different antigen selection regimes (*Pappas et al., 2014*) and may account for the low frequencies of bnAbs in natural repertoires (*Corti et al., 2017*). Characterizing the prevalence of these constraints on bnAb evolution may provide valuable insight for improving vaccination strategies (*Yewdell, 2013*; *Henry et al., 2018*).

To date, studies of antibody binding have been limited to small numbers of individual sequences, deep mutational scans of single mutations, and mutagenesis of small regions (*Pappas et al., 2014*; *Braden et al., 1998*; *Burks et al., 1997*; *Adams et al., 2016*; *Koenig et al., 2017*; *Forsyth et al., 2013*; *Wu et al., 2017*; *Xu et al., 2015*; *Madan et al., 2021*; *Schmidt et al., 2015*), due in part to practical constraints on library scale and the throughput of affinity assays. This has limited our ability to comprehensively characterize binding landscapes for naturally isolated bnAbs, which often involve many mutations spanning framework (FW) and complementarity-determining regions (CDR) (*Corti et al., 2017*; *Corti and Lanzavecchia, 2013*; *Klein et al., 2013*).

We overcome these challenges by generating combinatorially complete libraries of up to $\sim 10^5$ antibody sequences and assaying their binding affinities in a high-throughput yeast-display system (*Adams et al., 2016*). This approach enables us to infer the contributions of individual mutations as well as hundreds of pairwise and higher order interactions between mutations, revealing that these interactions can restrict evolutionary pathways leading to greater breadth. In particular, we find that mutational effects on binding affinity to diverse antigens display a nested structure, where increasingly large groups of specific mutations are required to gain affinity to divergent antigens, resulting in highly constrained paths to broad affinity. This pattern is not observed for more similar antigens, where many mutational paths to broad affinity are accessible. Further, these nested patterns of mutational effects provide new molecular insight into why sequential exposure to diverse antigens often favors greater breadth (*Wang et al., 2010*; *Krammer et al., 2012*; *Wang et al., 2015*; *Wang, 2017*; *Sachdeva et al., 2020*; *Molari et al., 2020*; *Sprenger et al., 2020*). Together, this work provides the first comprehensive characterization of antibody affinity landscapes and advances our understanding of the molecular constraints on bnAb evolution.

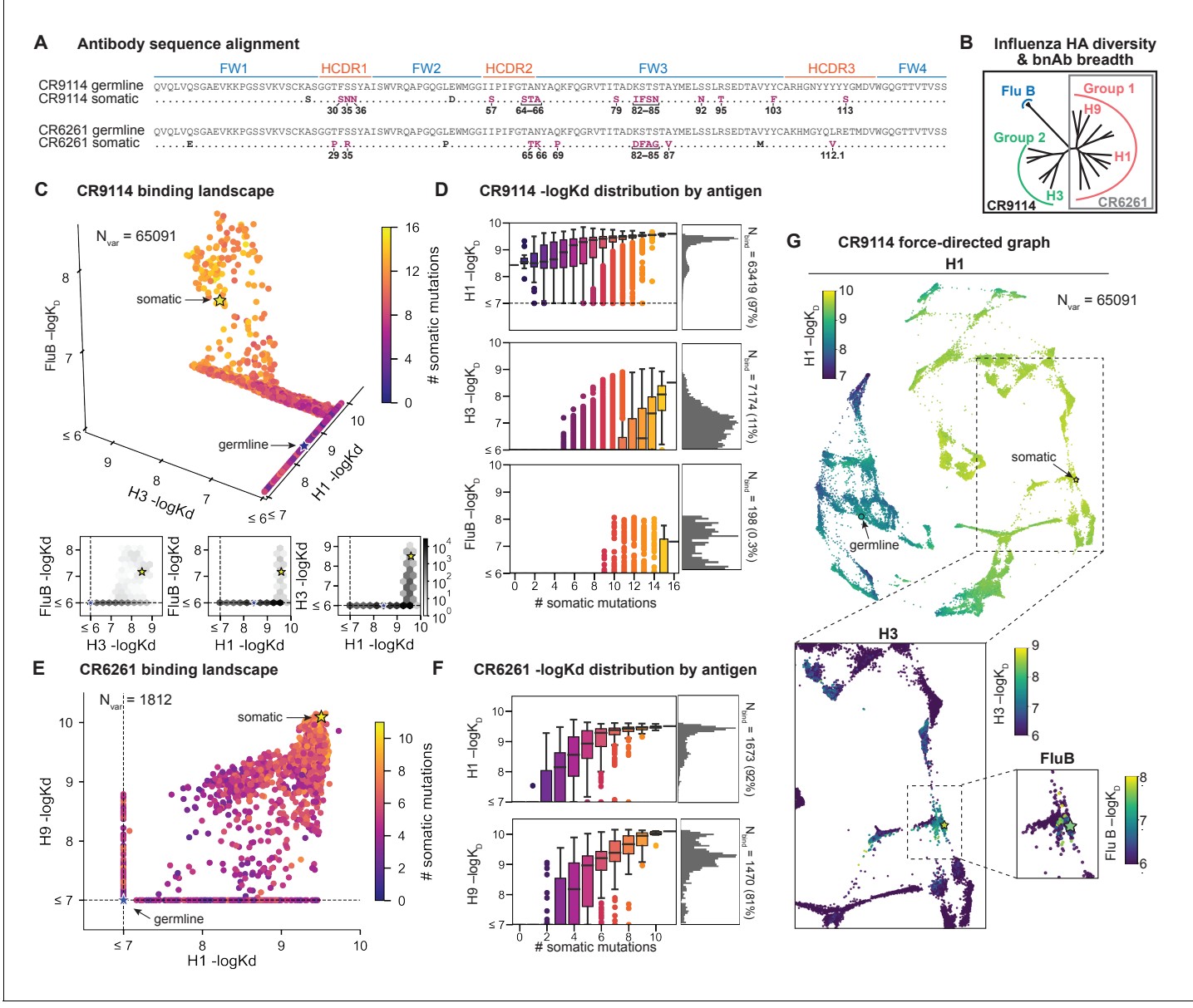

**Figure 1.** Binding landscapes. (**A**) Sequence alignment comparing somatic heavy chains to reconstructed germline sequences. Mutations under study (purple, numbered) and excluded mutations (black) are indicated; residues are numbered by IMGT unique numbering. (**B**) Influenza hemagglutinin phylogenetic tree with selected antigens and breadth of CR9114 (black box) and CR6261 (gray box) indicated. (**C, E**), Scatterplots of the (**C**) CR9114 library binding affinities against three antigens, with 2D planes shown below, and (**E**) CR6261 library binding affinities against two antigens. (**D, F**) Distributions of library binding affinities for (**D**) CR9114 and (**F**) CR6261 for each antigen (gray histogram, right) separated by number of somatic mutations (boxplots, left). Numbers and percentages of variants with measurable binding are indicated at right. (**G**), Force-directed graph of CR9114 H1 $-\log K_D$. Each variant (node) is connected to its 16 single-mutation neighbors (edges not shown for clarity); edges are weighted such that variants with similar genotypes and $-\log K_D$ tend to cluster. Nodes are colored by binding affinity to H1 (top; showing all 65,091 nodes), H3 (lower left inset; showing only the region containing nodes with $-\log K_D > 6$), and Flu B (lower right inset; showing only the region containing nodes with $-\log K_D > 6$).

The online version of this article includes the following source data and figure supplement(s) for figure 1:

**Source data 1.** CR9114 library -$\log K_D$ to H1, H3, and influenza B.

**Source data 2.** CR6261 library -$\log K_D$ to H1 and H9.

**Source data 3.** Isogenic flow cytometry measurements of $-\log K_D$ for select CR9114 and CR6261 variants.

**Figure supplement 1.** Experimental design and Tite-Seq workflow.

**Figure supplement 2.** Tite-Seq data quality.

**Figure supplement 3.** Antibody-antigen co-crystal structures.

**Figure supplement 4.** Force-directed graph for CR6261.

*Figure 1 continued on next page*

*Figure 1 continued*

**Figure supplement 5.** Expression of antibody libraries.
**Figure supplement 6.** Tite-Seq gating strategy.
**Figure supplement 7.** Reversions of excluded mutations.

## Results

### Binding affinity landscapes of CR9114 and CR6261

Here, we characterize the binding affinity landscapes of the two well-studied bnAbs noted above: CR9114 and CR6261. Specifically, we made all combinations of a set of mutations separating the germline and somatic sequences for CR9114 (16 mutations totaling 65,536 variants) and CR6261 (11 mutations totaling 2048 variants). These libraries include all heavy-chain mutations in these antibodies, except a few select mutations distant from the paratope (*Figure 1*, *Figure 1—figure supplement 7*, and see Materials and methods). Both antibodies engage antigens solely through their heavy-chain regions (*Dreyfus et al., 2012*; *Ekiert et al., 2009*), and thus are well-suited for yeast display as single-chain variable fragments (see Materials and methods) (*Boder and Wittrup, 1997*).

We use the Tite-Seq method (*Adams et al., 2016*), which integrates flow cytometry and sequencing (*Figure 1—figure supplement 1*), to assay equilibrium binding affinities of each scFv sequence in these libraries against select antigens that span the breadth of binding for each antibody (*Figure 1B*). For CR6261, we chose two divergent group 1 HA subtypes (H1 and H9; see *Figure 1—figure supplement 1*), while for CR9114, we chose the three highly divergent subtypes present in the vaccine (H1 from group 1, H3 from group 2, and influenza B; see *Figure 1—figure supplement 1*, *Throsby et al., 2008*). Inferred affinities outside our titration boundaries ($10^{-11}$–$10^{-6}$ M for H3 and influenza B, $10^{-12}$–$10^{-7}$ M for H1 and H9) are pinned to the boundary, as deviations beyond these boundaries are likely not physiologically relevant (*Batista and Neuberger, 1998*). Antibody expression is not strongly impacted by sequence identity, although some mutations have modest effects that may be inversely correlated with their effect on affinity (*Figure 1—figure supplement 5*). Affinities obtained by Tite-Seq are reproducible across biological triplicates (*Figure 1—figure supplement 2*; average standard error of 0.047 -log$K_D$ units across antibody-antigen pairs) and are highly accurate as verified for select variants by isogenic flow cytometry (*Figure 1—figure supplement 2*) and by solution-based affinity measurements made by others (*Throsby et al., 2008*; *Dreyfus et al., 2012*; *Lingwood et al., 2012*; *Pappas et al., 2014*).

We begin by examining the distribution of binding affinities across antigens for each antibody library (*Figure 1*). We observe that most CR9114 variants have measurable affinity to H1 (97%), fewer to H3 (11%), and still fewer to influenza B (0.3%) (*Figure 1C,D*). For H1, only a few mutations are needed to improve from the germline affinity. In contrast, variants are not able to bind H3 unless they have several more mutations, and many more for influenza B. This hierarchical structure is in striking contrast to the CR6261 library, in which most variants can bind both antigens (92% for H1, 81% for H9), variants have a similar $K_D$ distribution, and many variants display intermediate affinity to both antigens (*Figure 1E,F*). To visualize how genotypes give rise to the hierarchical structure of CR9114 binding affinities, we represent the binding affinities for H1 as a force-directed graph. Here, each variant is a node connected to its 16 single-mutation neighbors, with edge weights inversely proportional to the change in H1 binding affinity, such that variants with similar genotype and $K_D$ tend to form clusters (*Figure 1G*, *Figure 1—figure supplement 4*). Coloring this genotype-to-phenotype map by the –log$K_D$ to each of the three antigens, we see that sequences that bind H3 and influenza B are highly localized and overlapping, meaning that they share specific mutations. Thus, while many CR9114 variants strongly bind H1, only a specific subset bind multiple antigens.

### Mutational effects on binding to diverse antigens

To dissect how mutations drive the structure of these binding landscapes, we next infer specific mutational effects. We first log-transform binding affinities such that they are proportional to free energy changes ($\Delta G_{\text{binding}}$), which should combine additively under the natural null expectation (*Wells, 1990*; *Olson et al., 2014*). We then define a linear model with single mutational effects and interaction terms up to a specified order (defined relative to the unmutated germline sequence, see

Appendix 2 for alternatives), and fit coefficients by ordinary least squares regression. We use cross-validation to identify the maximal order of interaction for each antigen and report coefficients at each order from these best-fitting models (CR9114: fifth order for H1, fourth for H3, first for influenza B; CR6261: fourth order for H1 and H9; see Materials and methods). We note that the maximum order of interactions is affected by our inference power, particularly by the number of sequences with appreciable binding, and so we interpret these models as showing strong evidence of epistasis at least up to the order indicated. We explored the possibility of 'global' epistasis by inferring a nonlinear transformation of the $-logK_D$ values (*Sailer and Harms, 2017a*; *Otwinowski et al., 2018*), but found that this approach did not significantly reduce the order or number of specific interaction coefficients needed to explain the data (see Appendix 2). We also explored inferring epistasis up to full order using Walsh-Hadamard transformations; results are qualitatively similar but less conservative than cross-validated regression (see Appendix 2).

Examining the effect of individual mutations on the germline background (*Figure 2A,B*), we observe several mutations that enhance binding to all antigens (e.g. S83F for CR9114), and mutations that confer trade-offs for binding distinct antigens (e.g. F30S in CR9114 reduces affinity for H1 but enhances affinity for influenza B). Generally, large-effect mutations are at sites that contact HA (*Figure 2C*, *Figure 2—figure supplement 1*, *Dreyfus et al., 2012*; *Ekiert et al., 2009*). Consistent with prior biochemical and structural work, mutations essential for CR9114 breadth are spread throughout FW3 and the CDRs, forming hydrophobic contacts and hydrogen bonds with residues in the conserved HA stem epitope (*Dreyfus et al., 2012*; *Avnir et al., 2014*). We observe three specific mutations that are required for binding to H3 (present at over 90% frequency in the set of binding sequences), likely because they form hydrophobic contacts with HA (K82I and S83F) and reorient the CDR2 loop (I57S), which interacts with residues and a glycan in H3 that are distinct from those in H1 (*Dreyfus et al., 2012*). We also observe eight specific mutations that are required for binding to influenza B. Many of these breadth-conferring mutations are absent in CR6261, particularly those in CDR2 (*Dreyfus et al., 2012*; *Ekiert et al., 2009*). Notably, these sets of required mutations in CR9114 exhibit a nested structure: mutations beneficial for H1 are required for H3, and mutations required for H3 are required for influenza B, giving rise to the hierarchical structure of the binding landscape (*Figure 1C*).

Beyond these exceptionally synergistic interactions between required mutations, we find that epistasis is widespread, accounting for 18–33% of explained variance depending on the antibody-antigen pair (except influenza B, see Materials and methods, Appendix 2). Pairwise interactions are dominated by a few mutations (e.g. F30S for CR9114 and S35R for CR6261) that exhibit many interactions, both positive and negative, with other mutations (*Figure 2D,E*). Overall, mutations with strong pairwise interactions tend to be close in the crystal structure, although there are long-range pairwise interactions that are likely mediated by interactions with the antigen or conformational rearrangements (*Figure 2F*, *Figure 2—figure supplement 1*, *Dreyfus et al., 2012*; *Ekiert et al., 2009*; *Avnir et al., 2014*).

## High-order epistasis is dominated by a subset of mutations

Our dataset also allows us to resolve higher order epistasis. In addition to the required mutations, our models identify numerous strong third to fifth order interactions, with a subset of mutations participating in many mutual interactions at all orders. For CR9114 binding to H1, this subset consists of five mutations, distributed across three different regions of the heavy chain (*Figure 3A,B*). Some of these mutations likely generate (K82I, S83F) or abrogate (F30S) contacts to HA, and others (I57S, A65T) may indirectly impact HA binding by reorienting contact residues in CDR2 (*Dreyfus et al., 2012*; *Avnir et al., 2014*). Within this set of five residues, we first illustrate two examples of third-order epistasis by grouping sequences by their genotypes at these five sites (*Figure 3C*). Intriguingly, some mutations that are deleterious in the germline background ('−' annotations) are beneficial in doubly-mutated backgrounds ('+' annotations). For example, mutation F30S is significantly less deleterious in backgrounds with S83F than in the germline background, suggesting that new hydrophobic contacts in FW3 may be able to compensate for the potential loss of contacts in CDR1. Yet F30S unexpectedly becomes beneficial after an additional mutation I57S in CDR2, indicating more complex interactions between flexible CDR and FW loop regions (*Figure 3B,C*; *Dreyfus et al., 2012*).

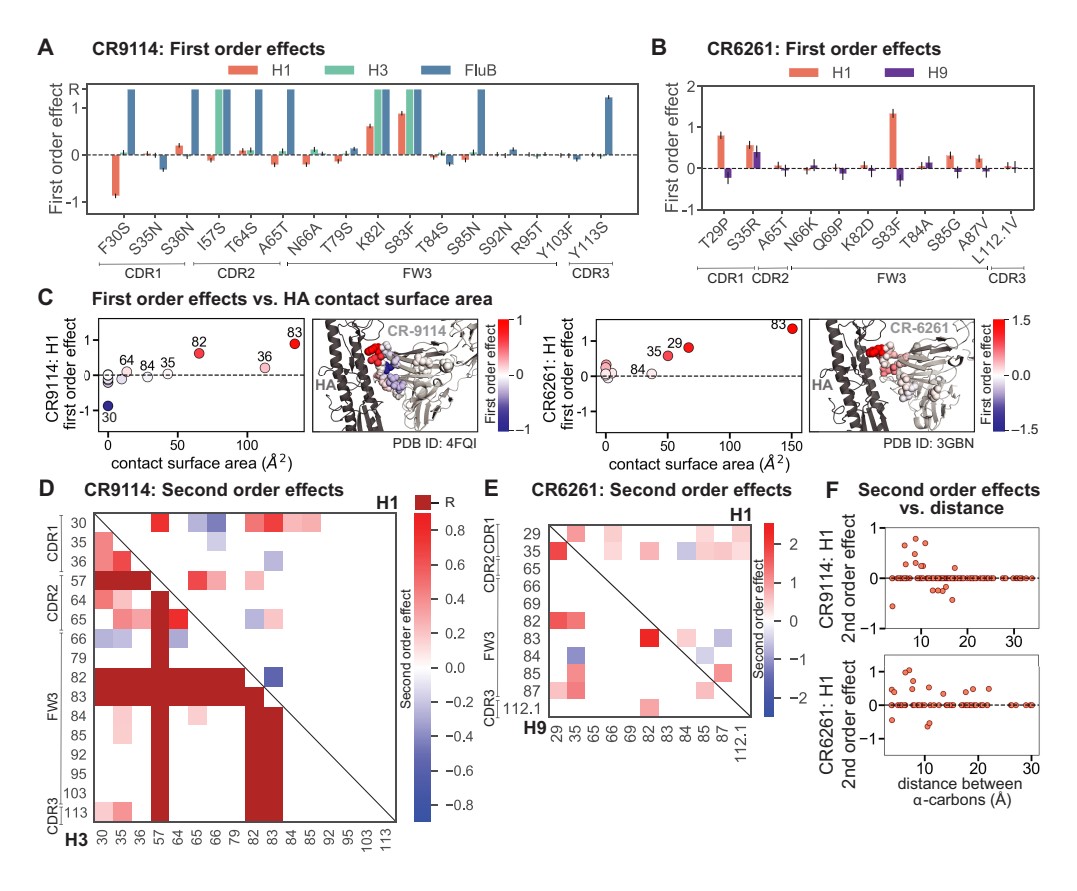

**Figure 2.** First- and second-order effects. (A, B) First-order effects inferred in best-fitting epistatic interaction models for (A) CR9114 and (B) CR6261. Mutations required for binding (present at over 90% frequency in binding sequences) have effect sizes denoted as 'R' and are removed from inference. Error bars indicate standard error. (C) First order effects for each site plotted against the contact surface area between the corresponding somatic residue and HA (left, CR9114; right, CR6261). Sites with notable contact area or effect size are labeled. Cocrystal structures are also shown; mutations are colored by first-order effect size. (D) Significant second-order epistatic interaction coefficients for CR9114 mutations (bottom left, H3; top right, H1). Interactions involving required mutations are shown in dark red. (E) Significant second-order coefficients for CR6261 mutations (bottom left, H9; top right, H1). Significance in (D), (E) indicates Bonferroni-corrected p-value < 0.05, see Materials and methods. (F) Second-order coefficients for H1 –logK$_D$ plotted against the distance between the respective $\alpha$-carbons in the crystal structures.

The online version of this article includes the following source data and figure supplement(s) for figure 2:

**Source data 1.** Interaction model coefficients for CR9114.

**Source data 2.** Interaction model coefficients for CR6261.

**Source data 3.** Tabulated contact surface area, number of HA contacts, and pairwise distances for mutations in CR9114 and CR6261.

**Figure supplement 1.** Structural context of first and second order effects.

To see how these high-order interactions drive the overall structure of the binding affinity landscape, we return to the force-directed graph, now colored by genotype at these five key sites (*Figure 3D*; only points corresponding to genotypes shown in *Figure 3C* are colored). We see that these five sites largely determine the overall structure of the map: points of the same color tend to cluster together, despite varying in their genotypes at the other 11 sites. However, we observe that interactions with other mutations do exist, as evidenced by separate clusters with the same color (e.g. the two clusters in teal for 57,65 are distinguished by a positive third-order interaction with site 64, *Figure 3E*). These patterns are not confined to the genotypes shown in *Figure 3C*; if we color all 32 possible genotypes at the five key sites, we observe the same general patterns (*Figure 3—figure supplement 1*; an interactive data browser for exploring these patterns of epistasis in CR9114 is available at: https://yodabrowser.netlify.app/yoda_browser/). Interactions between these five sites are also enriched for significant epistatic coefficients ($p<10^{-3}$; 26 of 31 possible terms are significant,

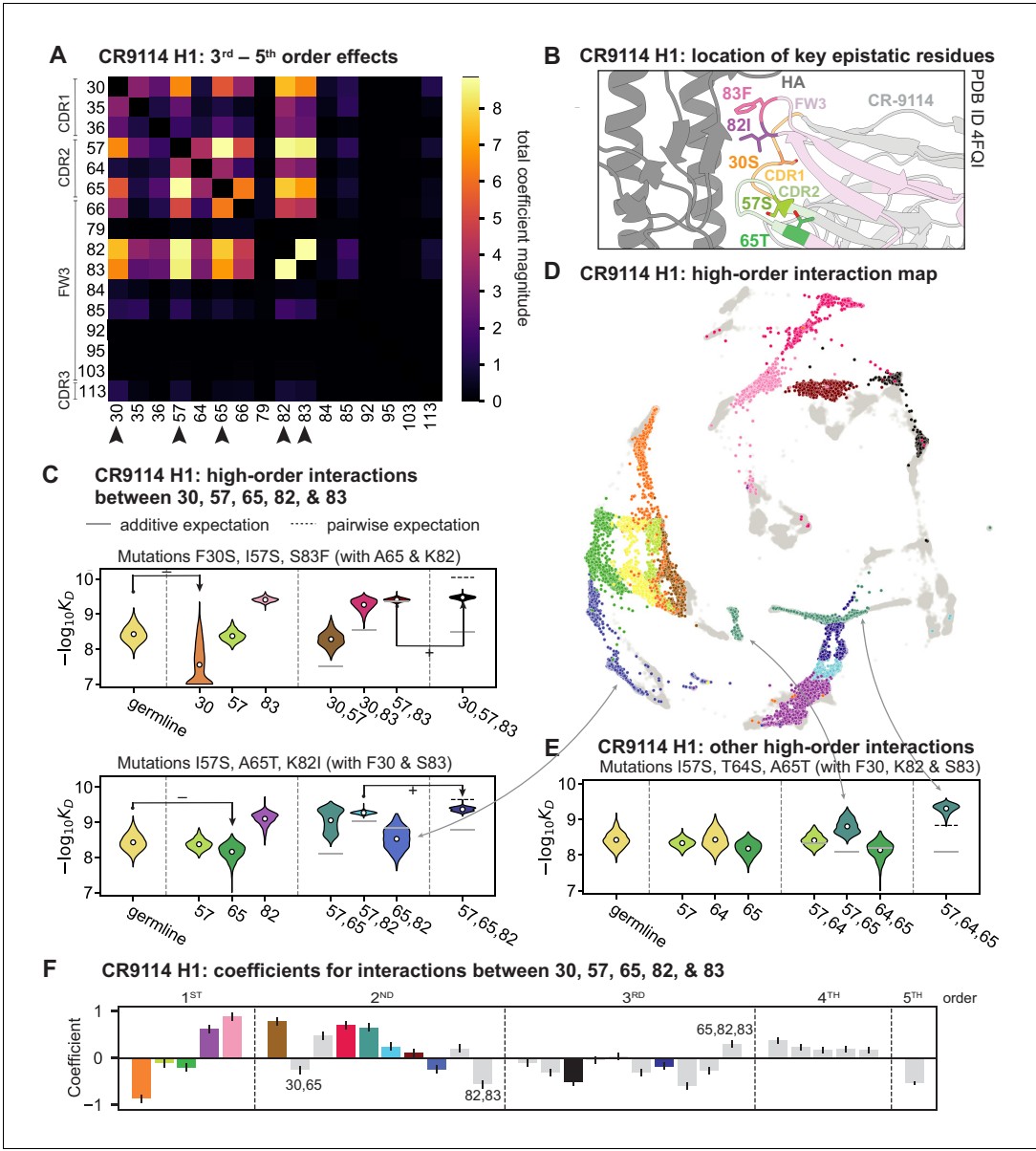

**Figure 3.** High-order epistasis for CR9114. (**A**) Total higher order epistatic contributions of CR9114 mutation pairs for binding H1. Color bar indicates the sum of absolute values of significant higher order interaction coefficients involving each pair of mutations; key epistatic residues indicated by arrows. Significance is given by Bonferroni-corrected p-value < 0.05, see Materials and methods. (**B**) Location of key epistatic residues in the CR9114 –HA co-crystal structure colored by region. (**C**) –logK$_D$ distributions for genotypes grouped by their identity at the five residues indicated in (**A**), (**B**), with means indicated as white dots ($N = 8,192$ genotypes per violin). Annotations indicate notable deleterious ('−') and beneficial ('+') mutational effects. (**D**) CR9114 force-directed graph from *Figure 1G*, colored as in (**C**) by the genotype at the five sites indicated in (**A**), (**B**). Genotypes not shown in (**C**) are shown in light gray. Data are also available in an interactive data browser at https://yodabrowser.netlify.app/yoda_browser/. (**E**), Third-order interaction involving site 64 accounts for distinct clusters (teal) corresponding to genotypes with mutations 57 and 65 in (**D**). Colors correspond to mutation groups in (**C**), (**D**) ($N = 4,096$ genotypes per violin). (**F**), Epistatic interaction coefficients among the five key sites from (**A**), (**B**). Colors for certain groups as in (**C**), (**D**); other groups denoted in gray, with notable terms labeled.

The online version of this article includes the following figure supplement(s) for figure 3:

**Figure supplement 1.** CR9114: interactions between five key sites.

**Figure supplement 2.** CR9114: interactions between other sets of five sites.

**Figure supplement 3.** High-order epistasis for CR9114 binding to H3.

compared to an average of 4 terms among all sets of five sites, *Figure 3—figure supplement 1*), including the fifth order interaction between all five residues (*Figure 3F*). Remarkably, these five mutations underlie significant high-order epistasis for other antigens as well: all five are either required for binding or participate extensively in interactions for H3 and influenza B (*Figure 3—figure supplement 3*).

Higher order epistasis in CR6261 is similarly dominated by a subset of mutations in CDR1 and FW3, at identical or neighboring positions as some key sites for CR9114 (*Figure 4A*). These mutations exhibit strong diminishing returns epistasis at third and fourth order, countering their synergistic pairwise effects, in a similar manner across both antigens (*Figure 4B*, *Figure 4—figure supplement 1*, *Figure 4—figure supplement 2*). Many fourth-order combinations of these mutations display interaction coefficients of similar magnitude (*Figure 4—figure supplement 1*), though they may be signatures of even higher order interactions that we are underpowered to infer.

A common approach to quantify how epistasis constrains mutational trajectories is to count 'uphill' paths (i.e. where affinity improves at every mutational step from the germline to the somatic sequence). We find that only a small fraction of potential paths are uphill (0.00005% +/- 0.00004% for CR9114 binding H1, and 0.2% +/- 0.04% for CR6261 binding H1, as estimated by bootstrap, see Materials and methods). However, we note that for all antibody-antigen combinations, the somatic sequence is not the global maximum of the landscape (the best-binding sequence) and some mutations have deleterious effects on average. Hence, strictly uphill paths are only possible due to sign epistasis, where normally deleterious mutations have beneficial effects in specific genetic backgrounds.

Overall, we see that mutational effects and interactions between them explain the affinity landscapes we observe. For CR9114, binding affinity to H1 can be achieved through different sets of few mutations with complex interactions. In contrast, a specific set of many mutations with strong synergistic interactions is required to bind H3, and to an even greater extent, influenza B (*Figure 2A*), giving rise to the landscape's hierarchical structure (*Figure 1C*). For CR6261, the higher order interactions are more similar between H1 and H9, which is consistent with the more correlated patterns of binding affinities between these two antigens (*Figure 1E*).

## Affinity to diverse antigens was likely acquired sequentially

The hierarchical nature of the CR9114 landscape suggests that this lineage developed affinity to each antigen sequentially. Considering the maximum –logK$_D$ achieved by sequences with a given

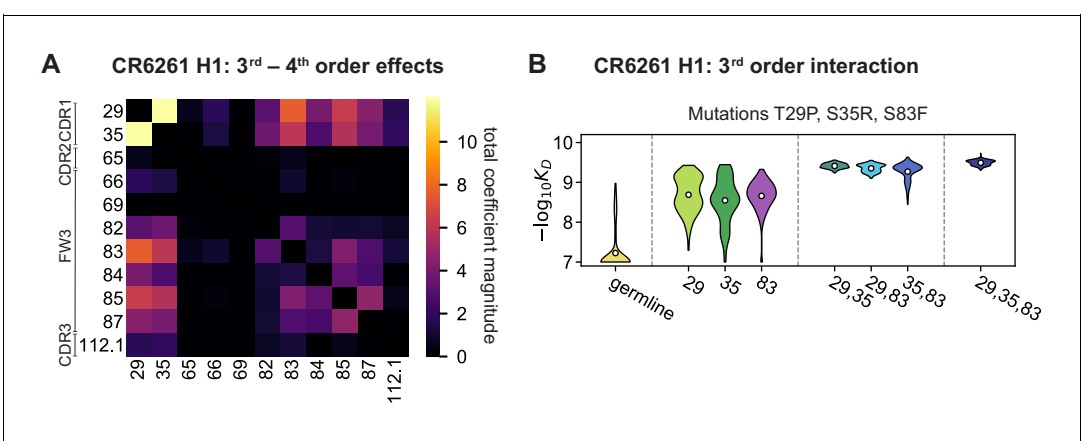

**Figure 4.** High-order epistasis for CR6261. (**A**) Total significant epistatic contributions of CR6261 mutation pairs for binding H1, as in *Figure 3A*. Significance is given by Bonferroni-corrected p-value < 0.05, see Materials and methods. (**B**) Third-order interaction for CR6261 H1 binding between mutations T29P, S35R, and S83F ($N = 256$ genotypes per violin).

The online version of this article includes the following figure supplement(s) for figure 4:

**Figure supplement 1.** CR6261: interactions between four sites.

**Figure supplement 2.** High-order epistasis for CR6261 binding to H9.

number of mutations (a proxy for time), we see that improvements in H1 binding can be realized early on, whereas improvements in H3 binding are not possible until later, and even later for influenza B (*Figure 5A*). In fact, the nested structure of affinity-enhancing mutations forces improvements in binding affinity to occur sequentially. If selection pressures were also experienced in this sequence, mutations that improve binding to the current antigen would lead to the genotypes required to begin improving binding to the next. Indeed, we find that for CR9114, there are more uphill paths leading to the somatic sequence if selection acts first on binding to H1 and later to H3 and influenza B (*Figure 5C*). In contrast, for CR6261, improvements in binding can occur early on for

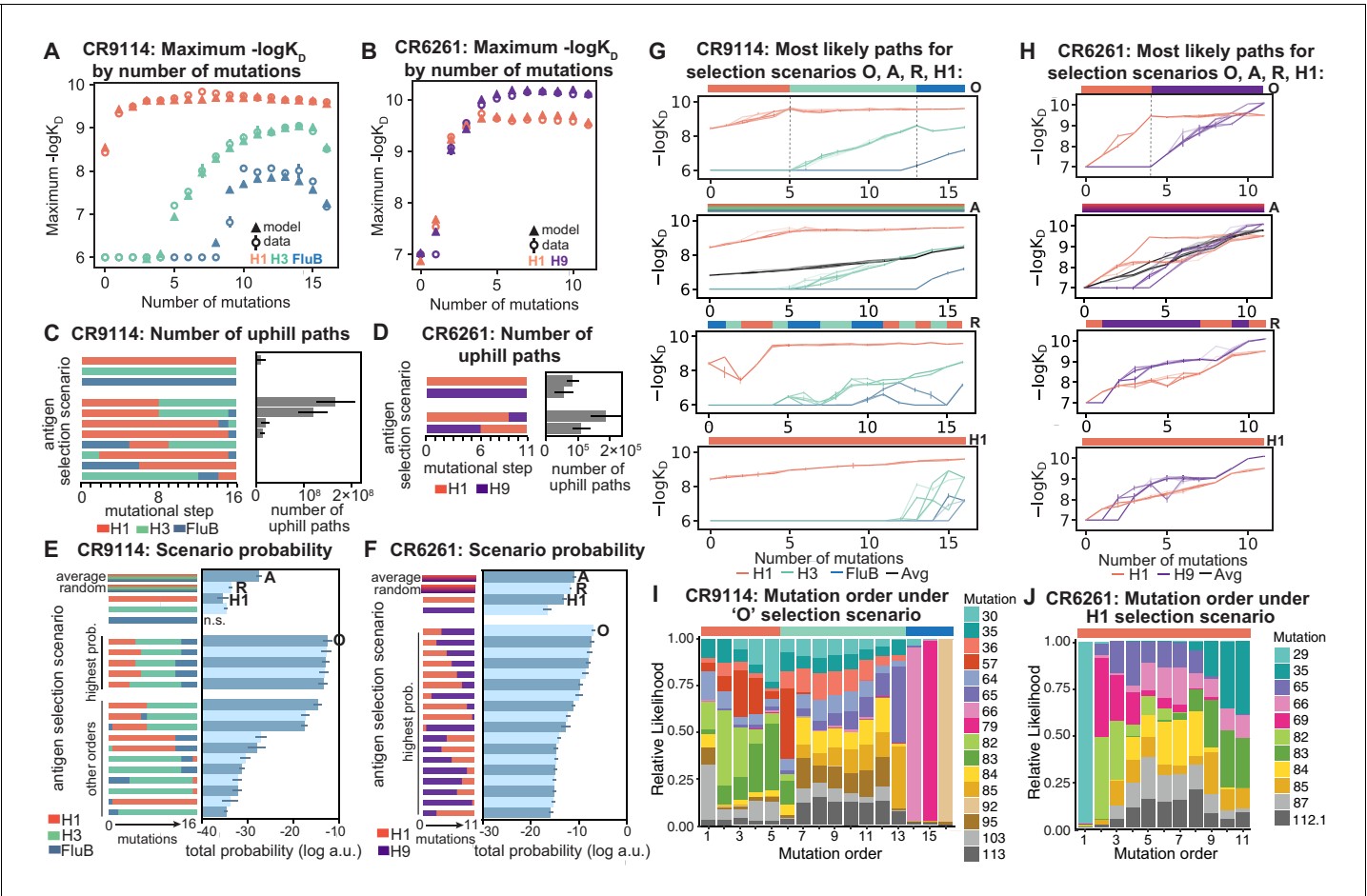

**Figure 5.** Antigen selection scenarios and likely mutational pathways. (A, B) Maximum binding affinity achievable for sequences with a given number of mutations. For each antigen for (A) CR9114 and (B) CR6261, the maximum observed (circles) and model-predicted (triangles) affinity for each number of somatic mutations is shown. (C, D) Total number of 'uphill' paths for select antigen selection scenarios (colored bars) for (C) CR9114 and (D) CR6261. Error bars indicate standard error obtained through bootstrap, see Materials and methods. (E, F) Total log probability (in arbitrary units) of mutational trajectories from germline to somatic sequence for (E) CR9114 and (F) CR6261 under different antigen selection scenarios, in a moderate selection model. Error bars indicate standard error obtained through bootstrap, see Materials and methods. (G, H) 25 most likely paths for (G) CR9114 and (H) CR6261, from select scenarios in (E, F); –logK$_D$ plotted for each antigen. For the random mixed scenario ('R'), a representative case is shown. 'A' indicates the average mixed scenario; 'O' indicates the optimal scenario. (I, J) Probability of mutation order under optimal antigen selection scenario 'O' for CR9114 (I) and H1 for CR6261 (J). Selection scenarios are as in (E, F) and shown in colored bar at top; the total probability (through all possible paths) for each mutation to occur at each mutational step is shown as stacked colored bars.

The online version of this article includes the following source data and figure supplement(s) for figure 5:

**Source data 1.** Total probability of mutational trajectories for CR9114 under different antigen selection scenarios.

**Source data 2.** Total probability of mutational trajectories for CR6261 under different antigen selection scenarios.

**Figure supplement 1.** Selection models.

**Figure supplement 2.** Variant probabilities for CR9114 under the optimal ('O') selection model.

**Figure supplement 3.** Probability of mutation order assuming moderate selection, under other antigen selection scenarios.

both antigens (*Figure 5B*) and the number of uphill paths is more similar across single-antigen and sequential selection pressures (*Figure 5D*).

To compare antigen selection scenarios more generally, we developed a framework that evaluates the total probability of all possible mutational pathways from germline to somatic, under an array of antigen selection scenarios (individual, sequential, and mixed). Our framework assumes that the probability of any mutational step is higher if $-logK_D$ increases, but does not necessarily forbid neutral or deleterious steps; we evaluate a variety of specific forms of this step probability and find that our major results are consistent (*Figure 5—figure supplement 1A*, see Materials and methods). We assume that each amino acid substitution occurs in a single mutational step; though there are amino acid substitutions that must proceed by multiple nucleotide mutations that may occur in a single round, or over multiple rounds, of somatic hypermutation (*Spisak et al., 2020*; *Unniraman and Schatz, 2007*). Mixed antigen regimes approximate exposure to a cocktail of antigens. We model these with two approaches: (1) 'average', using the average $-logK_D$ across all antigens, and (2) 'random', using $-logK_D$ for a randomly selected antigen at each step (note that using the maximum $-logK_D$ across antigens would always be trivially favored) (*Wang et al., 2015*). While these models simplify the complexities of affinity maturation *in vivo* (*Victora and Nussenzweig, 2012*), especially how affinity relates to B cell lineage dynamics and the mutational bias at the nucleotide level (*Spisak et al., 2020*), they provide insight into the relative probabilities of mutational paths under distinct antigen selection scenarios.

Again we find that the vast majority of likely antigen selection scenarios for CR9114 involve first H1, followed by H3, followed by influenza B (*Figure 5E*, *Figure 5—figure supplement 1B*). These results are underscored by examining improvement in $-logK_D$ along the most likely mutational paths for each scenario (*Figure 5G*): in the optimal sequential scenario, $-logK_D$ can improve substantially for each antigen in turn, while in an H1-only scenario, the improvements in H1 binding at each step are much more gradual, reducing the likelihood. The average mixed scenario shows qualitatively similar paths to the optimal sequential scenario, although with lower overall probability. In the random mixed scenario, even the best pathways are often unable to improve affinity to the randomly selected antigen, and affinity to antigens not under selection often declines, making these scenarios much less likely.

Given the optimal sequential selection scenario, the vast majority of genotypes are unlikely evolutionary intermediates to the somatic sequence (*Figure 5—figure supplement 2*). We visualize the impact of epistasis on mutational order by considering the probability of each mutation to occur at each mutational step (*Figure 5I*; *Figure 5—figure supplement 3*). The three antigen exposure epochs exhibit clear differences in favored mutations. Mutations I57S, K82I, and S83F must occur early, due to their strong synergistic interactions for all three antigens. In addition, we see that F30S is unlikely to happen very early (due to its sign epistasis under H1 selection) as well as unlikely to happen very late (due to its strong benefit under influenza B selection).

In contrast, for CR6261, all selection scenarios have relatively similar likelihood (*Figure 5F*, *Figure 5—figure supplement 1C*). Among sequential scenarios, however, those beginning with H1 are more likely than those beginning with H9, as the first two mutational steps can improve affinity to H1 more than H9, and mutations late in maturation can improve affinity to H9 more than H1 (*Figure 1F*, *Figure 5B*). Still, unlike CR9114, in both single antigen and mixed scenarios, there are many likely paths that continually improve in binding to both antigens (*Figure 5H*). Initially the order of mutations is highly constrained due to strong synergistic epistasis, and differences between selection scenarios reflect differences in mutational effects between antigens (*Figure 5J*, *Figure 5—figure supplement 3*). We note that T29P is highly likely to occur first in scenarios that begin with H1, as this is the only single mutation that can improve H1 affinity, albeit rather modestly.

## Discussion

Overall, we find that evolutionary pathways to bnAbs can be highly contingent on epistatic and pleiotropic effects of mutations. Specifically, the acquisition of breadth for CR9114 is extremely constrained and is likely to have occurred through exposure to diverse antigens in a specific order, due to the structure of correlations and interactions between mutational effects. In contrast, CR6261 could have acquired affinity to H1 and H9 in a continuous and simultaneous manner, perhaps

because these antigens are more similar; since H9 is not a commonly circulating strain, this breadth was likely acquired by chance (*Pappas et al., 2014*).

We note that we cannot conclusively determine how CR9114 and CR6261 evolved *in vivo*. The isolation of these specific antibodies from phage display libraries (*Throsby et al., 2008*; *Dreyfus et al., 2012*) was likely biased by the HA subtypes used for screening, and although unlikely, may have introduced mutations during PCR amplification. Regardless, these antibody sequences occupy regions of sequence space that are useful for understanding the relationship between sequence, affinity, and breadth. By characterizing their binding landscapes, we find that epistasis and trade-offs constrain the mutational pathways to these specific somatic sequences and their associated breadth. Indeed, we find that not all of the observed mutations are required to confer broad affinity, and future work is needed to explore what alternative pathways to breadth might be accessible through other mutations. It is also worth noting that selection pressure to bind the HA stem epitope on virions may be different from pressure to bind soluble recombinant HA, although several studies have found anti-stem antibody affinity to recombinant HA to be indicative of viral neutralization (*Dreyfus et al., 2012*; *Corti et al., 2011*; *Lingwood et al., 2012*). Further, stem-targeting bnAbs and their germline precursors have been characterized as polyreactive (*Bajic et al., 2019*; *Guthmiller et al., 2020*) and thus likely experience additional selection pressures that are not captured by our measurements and models, such as negative selection against autoreactivity. Although we cannot determine which specific antigens were involved in the selection of these antibodies *in vivo*, the diverse HA subtypes we employ capture variation representative of circulating influenza strains and thus serve as useful probes of varying levels of breadth (*Corti et al., 2017*). Future work integrating these measurements of affinity and breadth with measurements of stability and polyreactivity will provide important insight into the molecular constraints of bnAb evolution.

Notably, the landscapes characterized here are among the largest combinatorially complete collections of mutations published to date. In some respects, our observations of high-order interactions are consistent with earlier work in other proteins. In particular, epistasis has been found to affect function and constrain evolutionarily accessible pathways across functionally and structurally distinct proteins (*Weinreich et al., 2006*; *Starr et al., 2017*; *Ortlund et al., 2007*; *Podgornaia and Laub, 2015*; *Gong et al., 2013*; *Sailer and Harms, 2017b*; *Miton and Tokuriki, 2016*; *Poelwijk et al., 2019*; *Bank et al., 2015*). Further, pairwise and high-order epistasis appear to be common features of binding interfaces, such as enzyme-substrate and receptor-ligand interactions (*Weinreich et al., 2006*; *Starr et al., 2017*; *Ortlund et al., 2007*; *Podgornaia and Laub, 2015*; *Sailer and Harms, 2017b*; *Miton and Tokuriki, 2016*), and interacting mutations are often spaced in both sequence and structure, underscoring the complexity of protein-protein interfaces (*Podgornaia and Laub, 2015*; *Adams et al., 2019*; *Braden et al., 1998*; *Esmaielbeiki et al., 2016*; *Rotem et al., 2018*). On the other hand, the strongly synergistic, nested mutations crucial for CR9114 breadth are unusual, perhaps due to the nature of antibody-antigen interfaces or to the unique dynamics of affinity maturation (*Victora and Nussenzweig, 2012*). Together, these observations suggest that interactions between multiple mutations, such as those we characterize here, could play a substantial role in affinity maturation and may contribute to the rarity of bnAbs in natural repertoires.

Our findings provide molecular insight into the emerging picture of how selection can elicit broad affinity, illustrated by a substantial recent body of work ranging from *in vivo* experimental approaches (*Krammer et al., 2012*; *Wang et al., 2010*) to quantitative modeling of immune system dynamics (*Wang et al., 2015*; *Wang, 2017*; *Sachdeva et al., 2020*; *Molari et al., 2020*; *Sprenger et al., 2020*). These diverse studies often find that mixed-antigen regimens are less effective than sequential regimens at eliciting bnAbs. Our results demonstrate that, at least in part, this may be due to the intrinsic structure of the mutational landscape, defined by the complex interactions of mutational effects across antigens. With more studies of binding landscapes for diverse antibodies, we could better understand how such features generalize between different germline sequences, somatic mutation profiles, and antigen molecules. These insights will be valuable for leveraging germline sequence data and antigen exposure information to predict, design, and elicit bnAbs for therapeutic and immunization applications.

# Materials and methods

**Key resources table**

| Reagent type (species) or resource | Designation | Source or reference | Identifiers | Additional information |
|---|---|---|---|---|
| Strain, strain background (*Saccharomyces cerevisiae*) | EBY100 | ATCC | Cat#:MYA-4941 | |
| Cell line (*Spodoptera frugiperda*) | Sf9 | ThermoFisher | Cat#:B82501 | Cell line for production of baculovirus |
| Cell line (*Trichoplusia ni*) | High-Five | ThermoFisher | Cat#:B85502 | Cell line for HA expression |
| Antibody | Anti-cMyc-FITC (Mouse monoclonal) | Miltenyi Biotec | Cat#:130-116-485 | FACS (1:50) |
| Recombinant DNA reagent | pCT302 (plasmid) | Addgene | Cat#:41845 | |
| Recombinant DNA reagent | pCT302_CR9114 _germline (plasmid) | This paper | | Plasmid map in ***Supplementary file 4*** |
| Recombinant DNA reagent | pCT302_CR9114 _somatic (plasmid) | This paper | | Plasmid map in ***Supplementary file 5*** |
| Recombinant DNA reagent | pCT302_CR6261 _germline (plasmid) | This paper | | Plasmid map in ***Supplementary file 6*** |
| Recombinant DNA reagent | pCT302_CR6261 _somatic (plasmid) | This paper | | Plasmid map in ***Supplementary file 7*** |
| Recombinant DNA reagent | pET21a-BirA (plasmid) | Addgene | Cat#:20857 | |
| Sequence-based reagent | CR9114 golden gate dsDNA fragments | IDT | | Sequences listed in ***Supplementary file 2*** |
| Sequence-based reagent | CR6261 Golden Gate primers | IDT | | Sequences listed in ***Supplementary file 3*** |
| Sequence-based reagent | Illumina sequencing primers | IDT | | Sequences listed in ***Supplementary file 1*** |
| Peptide, recombinant protein | Streptavidin-RPE | ThermoFisher | Cat#:S866 | FACS (1:100) |
| Peptide, recombinant protein | Biotinylated A/New Caledonia/99 (H1) ectodomain | This paper | | Plasmid sequence in ***Supplementary file 8*** |
| Peptide, recombinant protein | Biotinylated A/Hong Kong/99 (H9) ectodomain | This paper | | Plasmid sequence in ***Supplementary file 9*** |
| Peptide, recombinant protein | Biotinylated A/Wisconsin/05 (H3) ectodomain | This paper | | Plasmid sequence in ***Supplementary file 10*** |
| Peptide, recombinant protein | Biotinylated B/Ohio/05 (Flu B) ectodomain | This paper | | Plasmid sequence in ***Supplementary file 11*** |
| Commercial assay or kit | Bac-to-Bac Kit | ThermoFisher | Cat#:10359016 | |
| Commercial assay or kit | Zymo Yeast Plasmid Miniprep II | Zymo Research | Cat#:D2004 | |

*Continued on next page*

*Continued*

| Reagent type (species) or resource | Designation | Source or reference | Identifiers | Additional information |
|---|---|---|---|---|
| Software, algorithm | Custom code | This paper | | https://github.com/ klawrence26/bnab-landscapes (copy archived at swh:1:rev:61c1673a101 ea739d5b7e9b282f6 bcfad41d7e90, *Phillips, 2021*) |
| Software, algorithm | Interactive CR9114 data browser | This paper | | https://yodabrowser. netlify. app/yoda_browser/ |

## Antibody library production

### Germline sequence reconstructions

For CR9114, we obtained the somatic heavy chain nucleotide sequence from *Dreyfus et al., 2012* (GenBank JX213639.1) and reconstructed the germline nucleotide sequence using IMGT (*Giudicelli et al., 2006*) and IgBLAST (*Ye et al., 2013*). Both methods assigned the same V-gene and J-gene alleles (IGHV1-69*06 and IGHJ6*02), but there is ambiguity in the D-gene assignment and at the V-D junction, particularly at site 109. The preferred IMGT junction alignment assigns a mutation here, S109N, while a different junction alignment from IgBLAST does not. Because of the inherent difficulty of reconstructing mutations in the junction region, especially in antibodies with a short D region, we chose the alignment without the mutation at site 109. Our reconstructed germline nucleotide sequence is available in *Supplementary file 12*. We then took the resulting germline and somatic amino acid sequences, as shown in *Figure 1A*, and constructed new nucleotide sequences codon-optimized for yeast.

For CR6261, the somatic and reconstructed germline heavy chain amino acid sequences were published in *Lingwood et al., 2012*. We used these sequences, similarly constructing codon-optimized nucleotide sequences for expression in yeast. The original somatic nucleotide sequence is also available (GenBank HI919029.1).

We note that all antibody libraries and clonal strains were constructed using somatic forms of the light chain, as these antibodies were isolated by combinatorial phage display (*Throsby et al., 2008*; *Dreyfus et al., 2012*), and so it is not possible to infer the naturally paired germline light chain sequence. Additionally, the CR9114 and CR6261 light chains were previously determined not to impact binding (*Lingwood et al., 2012*; *Dreyfus et al., 2012*; *Ekiert et al., 2009*). The somatic light chain sequence for CR9114 was obtained from *Dreyfus et al., 2012* (GenBank JX213640.1), and that for CR6261 was obtained from *Throsby et al., 2008* (GenBank HI919031.1).

### Mutation selection

CR9114 contains a total of 18 amino acid substitutions between the somatic variant and the reconstructed germline sequence. However, a library of $2^{18} = 262,144$ variants would be costly and time-consuming to produce and assay via our methods. We therefore identified two mutations that were distant from antigen contacts in the crystal structure: A25S and E51D (*Dreyfus et al., 2012*). We measured binding affinities for somatic sequences with and without these two mutations, and found that these variants had comparable affinities for both H1 and H3 (*Figure 1—figure supplement 7*). Although these mutations may have some small impact on binding, especially in combination with others, excluding them allowed for a simpler cloning strategy and a more manageable library size.

Similar to the CR9114 library design, we reduced the number of mutations present in the CR6261 library by excluding three mutations that were distant from antigen contacts in the crystal structure: 6QE, L50P, and V101M (*Ekiert et al., 2009*). We validated the marginal contribution of these mutations to binding by measuring the binding affinities for the somatic sequence with and without these mutations reverted to the respective germline residue (*Figure 1—figure supplement 7*).

## Yeast display plasmid and strains

To generate clonal yeast display strains and libraries for CR9114, we cloned scFv constructs (V$_L$ -Ser (Gly$_4$Ser)$_5$-V$_H$-Myc) into the pCT302 plasmid (*Midelfort et al., 2004*) (kind gift from Dane Wittrup; Addgene, Watertown, MA, #41845). For the clonal CR9114 somatic and germline strains, gene blocks corresponding to the somatic or inferred germline sequences were cloned into pCT302 by Gibson Assembly (*Gibson et al., 2009*) (plasmid maps in *Supplementary files 4–5*). For producing the plasmid backbone required for Golden Gate library generation (described below), we removed an existing Bsa-I site from the pCT302 plasmid by site-directed mutagenesis (Agilent, Santa Clara, CA, #200521) and replaced the V$_H$ domain with the *ccdB* gene. To generate clonal yeast strains, Gibson Assembly products were transformed into electrocompetent DH10B *E. coli* cells, and the resulting plasmids were mini-prepped and Sanger sequenced. Following sequence confirmation, plasmids were transformed into EBY100 yeast cells (ATCC #MYA-4941) as described in the high-efficiency yeast transformation protocol (*Gietz and Schiestl, 2007*). Transformants were plated on SDCAA-agar (1.71 g/L YNB without amino acids and ammonium sulfate [Sigma-Aldrich #Y1251], 5 g/L ammonium sulfate [Sigma-Aldrich #A4418], 2% dextrose [VWR #90000–904], 5 g/L Bacto casamino acids [VWR #223050], 100 g/L ampicillin [VWR #V0339], 2% Difco Noble Agar [VWR #90000–774]) and incubated at 30°C for 48 hr, single colonies were restruck on SDCAA-agar and again incubated at 30°C for 48 hr, and the resulting clonal yeast strains were verified to have the construct of interest by colony PCR. Construction of the yeast libraries is described below. All yeast strains were grown to saturation in SDCAA at 30°C, supplemented with 5% glycerol, and stored at −80°C.

CR6261 clonal yeast display strains and libraries were generated in an identical manner to that of CR9114, except where noted below (see *Supplementary files 6–7* for plasmid maps corresponding to the germline and somatic sequences).

## Golden gate assembly

For CR9114, due to the number of mutations required and their positions along the heavy chain coding sequence, we designed a library cloning strategy using Golden Gate combinatorial assembly (*Engler et al., 2008*). We divided the heavy chain coding region into five roughly equal fragments, ranging from 79 to 85 bp and each containing between 1 and 5 mutations. We added BsaI sites and additional overhangs to both ends of each fragment sequence, with cut sites carefully chosen so that the five fragments will assemble uniquely in their proper order within the plasmid backbone. For each fragment with $n$ mutations, we then ordered $2^n$ individual DNA duplexes with each possible combination of mutations (ranging from 2 to 32 versions for each fragment, a total of 66 fragments) from IDT (Coralville, IA) (see *Supplementary file 2*). By pooling the versions of each fragment in equal volumes, then pooling the five fragment pools in equimolar ratios, we obtained a randomized fragment mix containing all $2^{16}$ sequences present at approximately equal frequencies.

In addition to the fragment mix, we prepared the plasmid backbone for the Golden Gate reaction. We created a version of the yeast display plasmid with the counter-selection marker *ccdB* in place of the heavy chain sequence, with flanking BsaI sites (see above). We performed Golden Gate cloning using BsaI-HFv2 (NEB, Ipswich, MA, #R3733) following the manufacturer recommended protocol, with a 5:1 molar ratio of the fragment insert pool to plasmid backbone.

We transformed the assembly mix into electrocompetent *E. coli* (DH10B) via electroporation in 10 x 50 µL cell aliquots. We recovered each transformation in 5 mL SOC (2% tryptone, 0.5% yeast extract, 10 mM NaCl, 2.5 mM KCl, 10 mM MgCl$_2$, 10 mM MgSO$_4$, 20 mM glucose) at 37°C for 1 hr, and then transferred each to 100 mL of molten LB (1% tryptone, 0.5% yeast extract, 1% NaCl) containing 0.3% SeaPrep agarose (VWR, Radnor, PA #12001–922) spread into a thin layer in a 1L baffled flask (about 1 cm deep). The mixture was allowed to set on ice for an hour, after which it was kept for 18 hr at 37C to allow for dispersed growth of colonies in 3D. We observed $\sim 3 \times 10^5$ colonies per aliquot, for a total of $\sim 3$ million transformants. After mixing the flasks by shaking for 1 hr, we pelleted the cells and prepared plasmid by standard midiprep (Zymo Research, Irvine, CA, D4201), from which we obtained >120 µg of purified plasmid.

For CR6261, we designed a library cloning strategy also using Golden Gate combinatorial assembly, but with fragments created by PCR instead of purchased. We divided the heavy chain coding region into three roughly equal fragments, each containing between 2 and 5 mutations. We designed these fragments such that the mutations they contain are close to the 3' or 5' ends and

can thus be easily incorporated by PCR. PCR primers included mutations, BsaI sites, and unique overhangs chosen so that the three fragments would assemble uniquely in their proper order within the plasmid backbone. For each version of the three fragments, we generated dsDNA by PCR (52 PCR reactions in total; see *Supplementary file 3* for primer sequences). By pooling all versions of each fragment in equal volumes, then pooling the three fragment pools in equimolar ratios, we obtain a randomized fragment mix that, when ligated in the Golden Gate reaction, produces all of the $2^{11}$ sequences present at approximately equal frequencies.

In addition to the fragment mix, we prepared the plasmid backbone for the Golden Gate reaction. We created a version of the yeast display plasmid with the counter-selection marker *ccdB* in place of the three-fragment sequence, with flanking BsaI sites. We performed Golden Gate cloning using BsaI-HFv2 (NEB #R3733) following the manufacturer recommended protocol, with a 7:1 molar ratio of fragment inserts to plasmid backbone.

The transformation of the CR6261 library into *E. coli* was conducted in a similar fashion to that of CR9114, except that 8x50 µL cell aliquots were transformed, and 600,000 colonies were pooled for plasmid midiprep.

## Yeast library production

We then transformed the CR9114 plasmid library into EBY100 cells by standard high-efficiency protocols (*Gietz and Schiestl, 2007*). We recovered transformants in molten SDCAA (1.71 g/L YNB without amino acids and ammonium sulfate (Sigma-Aldrich #Y1251), 5 g/L ammonium sulfate (Sigma-Aldrich, St. Louis, MO, #A4418), 2% dextrose (VWR #90000–904), 5 g/L Bacto casamino acids (VWR #223050), 100 g/L ampicillin (VWR # V0339)) containing 0.35% SeaPrep agarose (VWR #12001–922) spread into a thin layer (about 1 cm deep). The mixture was allowed to set on ice for an hour, after which it was kept for 48 hr at 30℃ to allow for dispersed growth of colonies in 3D. From five such flasks, we obtained ~700,000 colonies (>10 times the library diversity). After mixing the flasks thoroughly by shaking for 1 hr, we grew cells in 5 mL tubes of liquid SDCAA for five generations and froze the saturated culture in 1 mL aliquots with 5% glycerol.

The CR6261 yeast library was generated in a manner identical to that of CR9114, except that ~60,000 colonies were pooled due to the smaller library size.

## Isogenic strain production

In addition to the full library, for both CR9114 and CR6261 we assayed a small number of variants by low-throughput flow cytometry for Tite-Seq validation. Any individual variant in the library can be produced in the same manner as described above: we simply selected the DNA duplex fragments corresponding to each desired variant and set up an individual Golden Gate reaction. The resulting assembled plasmid was transformed into *E. coli*, mini-prepped, and transformed into EBY100 in the same manner as described above. We verified the sequence identity of each variant by Sanger sequencing the entire scFv sequence.

We also constructed isogenic strains for validation experiments with genotypes that are not present in the full library. For CR9114, to test the impact of excluding mutations A24S and E46D, we constructed a strain containing the remaining 16 somatic mutations by cloning a gene block of the corresponding $V_H$ sequence into the germline CR9114 pCT302 plasmid via Gibson Assembly (*Figure 1—figure supplement 7*). For CR6261, we similarly constructed a strain with the Q6E, L50P, and V101M mutations reverted.

## Antigen production
### Choice of HA antigens

CR9114 was isolated from pooled PBMC from three donors who had received the trivalent 2006 influenza vaccine (*Throsby et al., 2008*; *Dreyfus et al., 2012*), which contained A/New Caledonia/20/1999 (H1N1), A/Wisconsin/67/2005 (H3N2), and B/Malaysia/2506/2004 (Victoria lineage) (*Ekiert et al., 2011*). CR6261 was isolated from pooled PBMC from the same three donors, plus an additional seven donors who did not receive the vaccine (*Throsby et al., 2008*). Because PBMC were isolated only 7 days after vaccination, although it is possible that CR6261 and CR9114 matured in response to these specific antigens, it is more likely that the vaccine elicited memory recall of these antibodies (*Victora and Nussenzweig, 2012*). Here, we chose to measure binding affinities to

diverse antigens spanning the range of breadth for both CR9114 and CR6261. CR9114 neutralizes strains across influenza A (groups 1 and 2) and influenza B, so we measured affinities to one strain from each of these groups, and selected vaccine-like strains: A/New Caledonia/20/1999 (H1N1), A/Wisconsin/67/2005 (H3N2), and B/Ohio/1/2005 (Victoria lineage). CR6261 neutralizes strains across influenza A group 1, thus we measured affinities to two strains from distinct subtypes within group 1: A/New Caledonia/20/1999 (H1N1) and A/Hong Kong/1073/1999 (H9N2). We note that CR9114 indeed binds A/Hong Kong/1073/1999 (H9N2) (*Dreyfus et al., 2012*), but CR9114 variant affinities for this strain were not measured here, as we prioritized measurements to antigens that span the breadth of each antibody.

## HA cloning, expression, and purification

Trimeric hemagglutinin (HA) antigen was produced as previously described (*Ekiert et al., 2011*; *Dreyfus et al., 2012*; *Margine et al., 2013*). Briefly, the HA ectodomain (Influenza A: residues 11–329 of HA1 and 1–176 of HA2 (H3 numbering); Influenza B: residues 1–523) of Influenza A/New Caledonia/1999 H1, Influenza A/Hong Kong/1999 H9, Influenza A/Wisconsin/2005 H3, and Influenza B/Ohio/2005, with N-terminal gp67 signal peptide and C-terminal biotinylation site (GGGLNDIFEA QKIEWHE), thrombin cleavage site, trimerization domain and His$_6$ tag, were cloned into pFastbac (plasmid maps in *Supplementary files 8–11*). Recombinant bacmid was generated using the ThermoFisher Bac-to-Bac kit (ThermoFisher, Waltham, MA, #10359016). Sf9 cells (ThermoFisher #B82501, not authenticated but verified to be mycoplasma-negative) were then transfected (ThermoFisher #A38915, not authenticated but verified to be mycoplasma-negative) with the resulting bacmids, and P0 HA-baculovirus was harvested 7 days post-transfection by clarifying viral supernatant at 1000 x g for 10 min. HA-baculovirus was then amplified twice by successively infecting 187 million Sf9 cells with 100 µL of viral supernatant and incubating in a humidified incubator at 28°C for 12 days. To induce HA expression, 105 million High-Five cells (ThermoFisher #B85502) were resuspended with 15 mL P2 HA-baculovirus, incubated for 20 min at room temperature, and then transferred to a 1 L non-baffled flask with 200 mL Corning Express-Five media (ThermoFisher #10486025) supplemented with 18 mM L-glutamine (VWR #45000–676). Expression cultures were incubated in a shaking incubator at 28C and 110 rpm for 48 hr, after which HA-containing media was clarified by spinning first at 1000 x g for 5 min at 4°C, and then by spinning the resulting supernatant again at 4000 x g for 20 min at 4C. The clarified media was then dialyzed into PBS (VWR #45000–448) by performing 4 x 2 hr 10-fold buffer exchanges to remove metal chelators from culture media. Dialyzed media was then combined with 10 mL equilibrated NiNTA resin (ThermoFisher #R90101), gently shaken for 3 hr at 4°C, and loaded onto a column. The resin was washed first with 15 column volumes Wash Buffer 1 (50 mM Tris pH 8 at 4C, 300 mM KCl, 10 mM imidazole) and subsequently with 15 column volumes Wash Buffer 2 (50 mM Tris pH 8 at 4C, 300 mM KCl, 20 mM imidazole). HA was eluted from the resin after 10 min incubation with Elution Buffer (50 mM Tris pH 8 at 4°C, 300 mM KCl, 250 mM imidazole). HA was then buffer exchanged into PBS using 10 KDa Amicon Ultra Centrifugal Filters (Millipore Sigma, Burlington, MA #UFC901008) and concentrated to at least 1 mg/mL for downstream biotinylation.

## BirA expression and purification

BirA was expressed and purified as previously described (*Ekiert et al., 2011*). Briefly, pET21a-BirA expression plasmid (*Howarth et al., 2005*) (kind gift from Alice Ting; Addgene #20857) was transformed into BL21 (DE3). Transformed BL21 cells were grown in 4 L baffled flasks with 1 L low-salt LB medium (5 g/L NaCl, 5 g/L yeast extract (VWR #90000–722), 10 g/L tryptone (VWR #90000–286)) at 37°C to an OD (600 nm) of ~0.8. The culture was then moved into cold water to bring it to 23°C, IPTG was added to a final concentration of 1 mM, and the culture was incubated at 23C for ~16 hr. The culture was then harvested by centrifugation (3000 x g, 10 min), resuspended in 30 mL lysis buffer (50 mM Tris pH 8 at 4C, 300 mM KCl, 10 mM imidazole, EDTA-free protease inhibitor cocktail tablet (Millipore Sigma #4693159001)), lysed by sonication (Branson Sonifier 450), and shaken at 4°C for 30 min. Lysate was clarified by spinning at 25,000 x g for 1 hr, and then the supernatant was incubated with 5 mL NiNTA resin at 4°C for 3 hr with gentle shaking. The resin was pelleted by spinning at 500 x g for 5 min and washed twice by gentle shaking with 35 mL lysis buffer at 4°C for 30 min. Protein was eluted with 20 mL Elution Buffer (50 mM Tris pH 8 at 4°C, 300 mM KCl, 250 mM

imidazole), buffer exchanged into Storage Buffer (50 mM Tris pH 7.5 at 4°C, 200 mM KCl, 5% glycerol) using 10 KDa Amicon Ultra Centrifugal Filters (Millipore Sigma #UFC901008), flash frozen in liquid nitrogen, and stored in single-use aliquots at −80°C.

## Biotinylation and HA-biotin quality control

Purified hemagglutinin was biotinylated as previously described (*Fairhead and Howarth, 2015*; *Ekiert et al., 2011*). Briefly, 100 µL HA (> 1 mg/mL) was incubated with 0.5 µL 1 M MgCl₂, 2 µL 100 mM ATP, 0.5 µL 50 mM biotin, and 2.5 µL BirA (10 mg/mL). This was mixed by gentle pipetting and incubated at 30C with gentle rocking. After 1 hr incubation, equivalent amounts of ATP, BirA, and biotin were added to the reaction, which was incubated for an additional hour at 30°C. Following the 2 hr incubation, the 100 µL reaction was exchanged thrice into 15 mL PBS using a 50 KDa MWCO buffer exchange column (Millipore Sigma #UFC905008). The degree of biotinylation was then assessed by a streptavidin gel-shift assay, as previously described (*Fairhead and Howarth, 2015*). Briefly, 10-fold molar excess streptavidin (Millipore Sigma #189730) was added to 4 µg biotinylated HA and incubated at room temperature for 5 min prior to running on SDS-PAGE. Gels were transferred to nitrocellulose membranes and probed with mouse anti-His monoclonal antibodies (Thermo-Fisher #R930-25) and Goat-anti-mouse secondary antibodies (LiCor, Lincoln, NE, Cat#925–32210). HA was verified to be > 80% biotinylated by densitometry.

## Tite-Seq assays

Tite-Seq was performed essentially as previously described (*Adams et al., 2016*), with some modifications as detailed below. For each antibody-antigen pair, three replicate Tite-Seq assays were performed on different days.

### Induction of antibody expression

On day 1, yeast scFv libraries, as well as germline and somatic clonal strains, were thawed by inoculating 5 mL SDCAA (1.71 g/L YNB without amino acids and ammonium sulfate (Sigma-Aldrich #Y1251), 5 g/L ammonium sulfate (Sigma-Aldrich #A4418), 2% dextrose (VWR #90000–904), 5 g/L Bacto casamino acids (VWR #223050), 100 g/L ampicillin (VWR # V0339)) with 150 µL glycerol stock (saturated culture with 5% glycerol) and rotated at 30°C for 20 hr. On day 2, yeast cultures were back-diluted to OD600 = 0.2 in 5 mL SDCAA and rotated at 30°C for approximately 4 hr, or until reaching log phase (OD600 = 0.4–0.8). 1.5 mL log-phase cells were then pelleted, resuspended in 4 mL SGDCAA (1.71 g/L YNB without amino acids and ammonium sulfate (Sigma-Aldrich #Y1251), 5 g/L ammonium sulfate (Sigma-Aldrich #A4418), 0.2% dextrose (VWR #90000–904), 1.8% galactose (Sigma-Aldrich #G0625), 5 g/L Bacto casamino acids (VWR #223050), 100 g/L ampicillin (VWR #V0339)), and rotated at room temperature for 20–22 hr.

### Primary antigen labeling

On day 3, 20–22 hr post-induction, yeast cultures were pelleted, washed twice with 0.1% PBSA (VWR #45001–130; GoldBio, St. Louis, MO, #A-420–50), and resuspended to an OD600 of 1. A total of 700 µL of OD1 yeast cells were labeled with biotinylated HA at each of eleven antigen concentrations (half-log increments spanning 1 pM – 100 nM for H1 and H9, and 10 pM – 1 µM for H3 and influenza B, as well as no HA), with volumes adjusted such that the number of antigen molecules was in 10-fold excess of antibody molecules (assuming 50,000 scFv/cell). Yeast-HA mixtures were rocked at 4°C for 24 hr.

### Secondary labeling

On day 4, yeast-HA complexes were pelleted by spinning at 3000 x g for 10 min at 4°C, washed twice with 5% PBSA + 2 mM EDTA, and simultaneously labeled with Streptavidin-RPE (1:100, Thermo Fisher #S866) and anti-cMyc-FITC (1:50, Miltenyi Biotec, Somerville, MA, #130-116-485) at 4°C for 45 min. Following secondary labeling, yeast were washed twice with 5% PBSA + 2 mM EDTA, and left on ice in the dark until sorting.

## Sorting and recovery

Yeast were sorted on a BD FACS Aria Illu, equipped with 405 nm, 440 nm, 488 nm, 561 nm, and 635 nm lasers, and an 85 micron fixed nozzle. Prior to sorting, single-color controls were used to compensate for the minimal FITC overlap with PE. Single cells were gated by FSC vs SSC, and then this population was sorted either by expression (FITC) or by expression and binding (PE). For all sorts, at least ten-fold excess of the library diversity was sorted (~1.6 million cells for CR9114; ~500,000 cells for CR6261). For the expression sorts, singlets were sorted into eight equivalent FITC log-spaced gates. For the binding sorts, FITC-positive cells were sorted into 4 PE bins (the PE-negative population comprised bin 1, and the PE-positive population was split into three equivalent log-spaced bins 2–4; see *Figure 1—figure supplement 6*). Polypropylene collection tubes were coated and filled with 1 mL YPD supplemented with 1% BSA and placed on ice until recovery. Sorted cells were pelleted by spinning at 3000 x g for 10 min, and supernatant was removed by pipette to avoid disturbing the pellets. Pellets were then resuspended in 4 mL SDCAA, a small amount was plated on SDCAA-agar to quantify recovery efficiency, and cultures were rocked at 30°C until reaching late-log phase (OD600 = 0.6–1.2).

## Sequencing library preparation

A total of 1.5 mL of late-log yeast cultures were pelleted and scFv plasmid was extracted using Zymo Yeast Plasmid Miniprep II (Zymo Research # D2004), per the manufacturer's instructions, and eluted in 10 µL elution buffer. Heavy-chain amplicon sequencing libraries were prepared by a two-step PCR as previously described (*Nguyen Ba et al., 2019*). In the first PCR, unique molecular identifiers (UMI), inline indices, and partial Illumina adapters were appended to the heavy chain through 3–5 amplification cycles to minimize PCR amplification bias. In the second PCR, the remainder of the Illumina adapter and sample-specific Illumina i5 and i7 indices were appended through 35 amplification cycles (see *Supplementary file 1* for primer sequences). The first PCR used 5 µL plasmid DNA as template in a 25 µL reaction volume, with Q5 polymerase according to the manufactuer's instructions (NEB # M0491L), and was incubated in a thermocycler with the following program: 1. 60 s at 98°C, 2. 10 s at 98°C, 3. 30 s at 66°C, 4. 30 s at 72°C, 5. GOTO 2, 2-4x, 6. 60 s at 72°C. PCR products were then combined with carrier RNA and purified by 1.1X Aline beads (Aline Biosciences #C-1003–5), and eluted in 35 µL elution buffer. 33 µL of the elution was used as input for the second PCR, in a total volume of 50 µL using Kapa polymerase (Kapa Biosystems #KK2502) according to the manufacturer's instructions, and incubated in a thermocycler with the following program: 1. 30 s at 98°C, 2. 20 s at 98°C, 3. 30 s at 62°C, 4. 30 s at 72°C, 5. GOTO 2, 34x, 6. 300 s at 72°C. The resulting sequencing libraries were purified by 0.85X Aline beads, amplicon size was verified to be ~500 bp by running on a 1% agarose gel, and amplicon concentration was quantified by a fluorescent DNA-binding dye (Biotium, Fremont, CA, #31068, per manufacturer's instructions). Amplicons were then pooled for each gate according to the number of sorted cells to ensure even sequencing coverage. The pool was further size-selected by a two-sided Aline bead cleanup (0.55–0.85X), and the final pool size was verified by Tapestation 5000 HS and 1000 HS. Final sequencing library concentration was determined by Qubit fluorometer and sequenced on an Illumina NovaSeq S2 or Miseq v3 (2x150) with 5% PhiX.

## Sequencing data processing

We first processed our raw sequencing reads to identify and extract the indexes and mutational sites, discarding priming regions and the constant regions between mutations. To do so, we developed custom Python scripts using the approximate regular expression library regex (*Barnett, 2013*), which allowed us to handle complications in sequence parsing that arise from the irregular lengths of the indices and from sequencing errors. We accept sequences that match the entire read (with no restrictions on bases at mutational sites) within the following mismatch tolerances: two mismatches in the multiplexing index, two mismatches in the priming site, and 15 substitution mismatches within the 170 bases of constant antibody sequence.

We then examine the mutational sites to call germline or somatic alleles, producing binary genotypes ('0' for germline or '1' for somatic at each position). We require the exact germline or somatic sequence at every site: if there are any substitution errors in any of the mutation sites, the entire read is rejected. While it is possible to perform error correction based on Hamming distance to

rescue reads with a few substitution errors, we find that on average only <8% of reads per sample contain any errors, and so we adopt the conservative approach of requiring perfect matching.

We next discarded sequencing reads with any mismatched indices (four total indices from the two PCR reactions), as well as reads with duplicate UMI sequences. Counts for each genotype were then tabulated, producing the final counts used for binding affinity inference (see below). On average, across all antigens and replicates, we obtain a mean coverage of ~350 for CR9114 and ~950 for CR6261, and a median coverage of ~250 for CR9114 and ~900 for CR6261.

## Isogenic validation

Induction of scFv surface display, primary labeling, and secondary labeling of isogenic strains were performed identically to the Tite-Seq assay, except yeast cell and antigen volumes were scaled down by a factor of 10. Yeast cell FITC (scFv expression) and R-PE (HA binding) fluorescence intensity was assayed on a BD LSR Fortessa equipped with four lasers (440, 488, 561, and 633 nm). The equilibrium binding affinities ($K_D$) for each variant are inferred by fitting the log of a Hill function to the mean log R-PE fluorescence of scFv-expressing (FITC+) singlet yeast cells:

$$\text{mean log fluorescence} = \log_{10}\left(A_s \frac{c}{c + K_{D,s}} + B_s\right), \tag{1}$$

where $c$ is the antigen concentration in molar units, $A_s$ is the increase in fluorescence due to saturation with antigen, $B_s$ is the background fluorescence, and $K_{D,s}$ is the equilibrium binding affinity. All isogenic measurements were performed in two to three biological replicates; see *Figure 1—source data 3* for isogenic $-\log_{10} K_D$.

## **Tite-Seq binding affinity inference**

### Mean-bin approach

To infer binding affinities using a simple mean-bin approach (*Peterman and Levine, 2016*), we incorporate sequencing data (the unique read counts of each genotype sequence $s$ in bin $b$ at concentration $c$, $R_{b,s,c}$) with flow cytometry data (the mean and standard deviation of $\log_{10}$-fluorescence of sorted cells in each bin $b$ at concentration $c$, $F_{b,c}$ and $\sigma_{F_{b,c}}$ respectively, and cell counts for each bin $b$ at each concentration $c$, $C_{b,c}$).

The mean log-fluorescence of each genotype sequence at each of the 12 antigen concentrations is calculated as:

$$\overline{F}_{s,c} = \sum_b F_{b,c}\, p_{b,s|c}, \tag{2}$$

where $p_{b,s|c}$ is the probability a cell with sequence $s$ would be sorted into bin b at concentration c is estimated from the sequencing read counts as:

$$p_{b,s|c} = \frac{\frac{R_{b,s,c}}{\sum_{s'} R_{b,s',c}} \cdot C_{b,c}}{\sum_{b'}\left(\frac{R_{b',s,c}}{\sum_{s'} R_{b',s',c}} \cdot C_{b',c}\right)}, \tag{3}$$

in other words, the fraction of total reads in the bin corresponding to sequence $s$, scaled by the number of sorted cells in that bin, normalized over the four bins for each concentration.

The uncertainty in the mean bin inference was propagated as:

$$\delta\overline{F}_{s,c} = \sqrt{\sum_b \left(\delta F_{b,c}^2\, p_{b,s|c}^2 + F_{b,c}^2\, \delta p_{b,s|c}^2\right)}. \tag{4}$$

Here, $\delta F_{b,c}$ represents the spread in log-fluorescence values of cells sorted into the same bin $b$. While we could estimate this value using the bin width, in practice we find that the distribution of cell log-fluorescence values in a bin is far from uniform across the bin width. The distribution is often not normal either, but we find that approximating $\delta F_{b,c} \approx \sigma_{F_{b,c}}$, or the standard deviation in $\log_{10}$-fluorescence of cells sorted into bin $b$ at concentration $c$, adequately captures the typical variation. The

error in $p_{b,s|c}$ arises largely from the sampling process of sequencing, which can be approximated as a Poisson process when read counts are relatively high. This gives

$$\delta p_{b,s|c} = \frac{p_{b,s|c}}{\sqrt{R_{b,s,c}}}. \tag{5}$$

Thus, $\delta \overline{F}_{s,c}$ can be written as

$$\delta \overline{F}_{s,c} = \sqrt{\sum_b \left( \sigma^2_{F_{b,c}} p^2_{b,s|c} + F^2_{b,c} \frac{p^2_{b,s|c}}{R_{b,s,c}} \right)}. \tag{6}$$

The equilibrium binding affinities ($K_D$) for each variant are inferred by fitting the logarithm of a Hill function to the resulting mean $\log_{10}$-fluorescence across the twelve antigen concentrations:

$$\overline{F}_{s,c} = \log_{10} \left( A_s \frac{c}{c + K_{D,s}} + B_s \right), \tag{7}$$

where $c$ is the antigen concentration in molar units, $A_s$ is the increase in fluorescence due to saturation with antigen, $B_s$ is the background fluorescence, and $K_{D,s}$ is the binding affinity. Fitting was performed with the *curve_fit* function of the Python package *scipy.optimize*. Reasonable bounds on the values of A($10^3$-$10^5$), B($10^0$-$10^3$) and K$_D$($10^{-14}$-$10^{-5}$) were imposed. Sequences leading to a failed optimization were deemed 'non-binding'.

Inferred $K_D$ outside of the titration boundaries were then pinned to the boundaries ($10^{-12}$ and $10^{-7}$ for H1 and H9; $10^{-11}$ and $10^{-6}$ for H3 and FluB). Inferred $K_D$ with high error (standard deviation of $\log_{10} K_D > 1.0$) or resulting from a poor fit ($r^2 < 0.8$) were removed from the data set prior to averaging $-\log_{10} K_D$ values across biological replicates.

We also explored an alternative maximum-likelihood framework for inferring binding affinities (see Appendix 1), but found it to be less accurate than the mean-bin approach when compared to isogenic flow cytometry measurements. Thus, we restricted our analysis to the simpler and more robust mean-bin inference presented here.

## Force-directed layouts

To represent the high-dimensional binding affinity landscape in two dimensions, we use a force-directed graph layout approach. Each sequence in the antibody library is a node, connected by edges to its single-mutation neighbors (sequences that can be reached by one additional somatic mutation). An edge between two sequences $s$ and $t$ is given the weight

$$w_{s,t} = \frac{1}{0.01 + \left| \log_{10}(K^{\mathrm{ag}}_{D,s}) - \log_{10}(K^{\mathrm{ag}}_{D,t}) \right|}, \tag{8}$$

where $K^{\mathrm{ag}}_D$ represent binding affinities to a particular antigen, ag. In the layouts shown in the main text, we use binding affinities to H1 for both CR6261 and CR9114. In force-directed layouts, edge weights correspond to the effective spring constant that tends to pull nodes closer together. Thus, a mutation from sequence $s$ to $t$ that has little impact on binding will cause that edge weight to be large, and the nodes will be pulled strongly together. A mutation from sequence $s$ to $t$ that causes a large difference in binding affinity (positive or negative) to the antigen will reduce the edge weight, moving those nodes further apart. After assigning all edge weights, we use the layout function *layout_drl* from the Python package *iGraph*, with default settings, to obtain the layout coordinates for each variant.

## Expression data

As noted above, antibody libraries were sorted into eight bins along the FITC-A fluorescence axis (where FITC-A fluorescence is proportional to expression), each comprising 12.5% of the total singlet population (*Figure 1—figure supplement 6*). The mean expression log-fluorescence was computed for each variant using the corresponding variant counts and fluorescence data, as described above for the mean-bin $K_D$ inference. These expression values were then averaged across all biological

replicates for each antibody (nine replicates for CR9114, six replicates for CR6261), and correlation between biological replicates, as well as with $-\log_{10} K_D$ values, are illustrated in *Figure 1—figure supplement 5*. For the isogenic flow cytometry measurements, variant expression was computed as the mean log FITC-A fluorescence.

## Epistasis analysis
### Linear interaction models

To infer specific mutational effects, we begin with simple linear models where the effects of mutations (and mutation combinations) add to produce phenotypes. Our log-transformed phenotypes for each variant $s$, $y_s = -\log_{10}(K_{D,s})$, are proportional to free-energy changes, and thus a natural null expectation is that they combine additively (*Wells, 1990*; *Olson et al., 2014*) (although we also consider nonadditive epistatic interactions between individual loci here, and analyze the effects of an overall nonlinear transformation of this data in Appendix 2). Our additive-only model is

$$y_s = \beta_0 + \sum_{i=1}^{L} \beta_i x_{i,s} + \varepsilon, \tag{9}$$

where $L$ is the number of mutations for a given antibody, $\beta_0$ is an intercept term, $\beta_i$ is the effect of the mutation at site $i$, $x_{i,s}$ is the genotype of variant $s$ at site $i$, and $\varepsilon$ represents independently and identically distributed errors. Our general linear interaction models are

$$y_s = \beta_0 + \sum_{i} \beta_i x_{i,s} + \sum_{i<j}^{L} \beta_{ij} x_{i,s} x_{j,s} + \sum_{i<j<k}^{L} \beta_{ijk} x_{i,s} x_{j,s} x_{k,s} + \ldots + \varepsilon \tag{10}$$

where $\beta_{ij}$ represent second-order interaction coefficients between distinct sites $i$ and $j$, $\beta_{ijk}$ represent third-order interaction coefficients, and so on up to the desired maximum order of interaction.

There are multiple alternative coding systems for the binary genotypes $x_{i,s}$ that affect the values of inferred effects $\beta$ as well as their interpretation. Two common choices are (1) $x_{i,s} \in \{0, 1\}$, often called 'biochemical' or 'local' epistasis, and (2) $x_{i,s} \in \{-1, 1\}$, often called 'statistical' or 'ensemble' epistasis (*Poelwijk et al., 2016*). These frameworks are equivalent and related by a simple linear transformation, but the values of the coefficients vary between frameworks and have different interpretations. For ease of interpretation, in the Main Text and Figures we always show results obtained from inference in the biochemical epistasis framework. In Appendix 2, we discuss the differences between these two frameworks, and present results from inference in the statistical epistasis framework.

For an antibody with $L$ mutations, there are $L$ possible orders of interactions, with a total of $2^L$ epistatic coefficients $\beta$. From a measurement of $y$ for all $2^L$ possible sequences, there is a simple linear transformation to calculate the resulting $2^L$ parameters (*Poelwijk et al., 2016*). This is a simple and fast approach to the calculation of epistasis that is widely used (*Sailer and Harms, 2017b*; *Poelwijk et al., 2019*), and we explore this approach in Appendix 2. However, we may instead wish to restrict our model to a lower order and examine whether it can explain the data with far fewer than $2^L$ parameters, as a conservative approach to detecting high-order epistasis.

Specifically, we truncate the model above at a maximum order $n$ and then fit and evaluate the resulting model. We begin with $n = 1$ and continue to increase $n$ until the optimal model has been identified. There are multiple strategies for selecting between models with different numbers of parameters, such as AIC and BIC; here we take a cross-validation approach. For each fold, we hold out 10% of the dataset, train models at each maximum order on the remaining 90%, and evaluate the prediction performance ($R^2$) of the model on the held-out test set. After averaging the performances across all 10 folds for each truncated model, we choose the order that maximizes the test set performance as the optimal maximal order of interaction. We then re-train the model truncated at this order on the full dataset to obtain the final coefficients. We find that the optimal model identified by cross-validation for each antibody-antigen pair satisfies $p<N$ by ~1 order of magnitude, where $p$ is the total number of model coefficients and $N$ the number of data points with measurable binding affinity. This gives confidence that our parameter estimates are well constrained by the data, even in the absence of other regularization (such as Lasso or Ridge regularization approaches).

To train a model of given order on a set of sequences, we use ordinary least squares (OLS) regression with the Python package *statsmodels*. From this, we obtain the coefficient values $\beta$ with their standard errors and p-values. To define significance of coefficients, we use a p-value cutoff of 0.05 with Bonferroni correction by the total number of model parameters. Coefficients, standard errors, p-values, and Bonferroni-corrected 95% confidence intervals are reported in *Figure 1— source data 1* and *2*. We also predict phenotypes $\hat{y}$ for each sequence from the coefficients and use these values in *Figure 5A,B*.

For CR9114 binding to influenza B, the number of sequences used for inference is far fewer than other antibody-antigen pairs ($N = 256$), due to the large number of required mutations. We therefore use a fivefold rather than 10-fold split to reduce the test set noise. Nevertheless, the cross-validation procedure identifies a first-order (additive) model as optimal, due to the smaller sample size.

## Structural analysis of epistatic coefficients

To examine the structural context of linear and pairwise coefficients, we performed three simple analyses. (1) First, we used ChimeraX (*Pettersen et al., 2021*) to calculate the buried surface area between HA and each mutated residue in CR9114 and CR6261, using the measure buriedarea function and the default probeRadius of 1.4 angstroms to approximate a water molecule. We plot this 'contact surface area' vs the linear effect of the corresponding mutation on HA binding (*Figure 2C*; *Figure 2—figure supplement 1A*). (2) We used PyMol (*Schrodinger, LLC, 2015*) to count the number of HA residues within six angstroms of each antibody mutation site. Six angstroms was chosen as an upper limit to capture potential antibody-antigen interactions (*Bondi, 1964*; *Baker and Hubbard, 1984*; *Israelachvili and Pashley, 1982*; *Ekiert et al., 2009*; *Dreyfus et al., 2012*), although we note that this analysis is robust to other distance thresholds. (3) We also used PyMol to measure the distances between $\alpha$-carbons for all mutation pairs, and plotted these distances against the corresponding pairwise epistatic terms (*Figure 2F*; *Figure 2—figure supplement 1B*). We note that each of these analyses were performed with co-crystal structures of the somatic antibodies with HA (PDB ID: 4FQI (CR9114 –H5; CR9114 –H1 crystal structure not available) *Dreyfus et al., 2012*; 4FQY (CR9114 –H3) *Dreyfus et al., 2012*; 3GBN (CR6261 –H1) *Ekiert et al., 2009*).

## Pathway analysis

### Selection models

To study the likelihood of various mutational pathways leading from the germline to the somatic sequence, we must assume a selection model. Selection in germinal centers is considerably more complex than in classical population genetics models, involving spatial structure, changing population sizes, and T-cell-mediated selection, among other factors (*Mesin et al., 2016*). Capturing these aspects in quantitative models is an active field of research (*Amitai et al., 2017*). However, here we wish to adopt an extremely simple model of selection as a first step in understanding the impacts of the binding affinity landscape on antibody selection, with the goal of understanding the implications of the expectation that mutational steps become more probable as their effect on binding affinity becomes more positive. Combining the more realistic models of immune selection with our detailed characterization of mutational effects on antigen binding affinity remains an interesting avenue for future work.

Here, we restrict to the weak-mutation regime where mutation fixation events occur independently of one another. Selection proceeds as a Markov process, where the population is characterized by a single sequence that acquires a single mutation at each discrete step (*McCandlish, 2011*). We choose a simple form for the fixation probability of a mutation from sequence $s$ to sequence $t$, as discussed below. This then determines the transition probability for the population to move from $s$ to $t$. We assume that sequences cannot back-mutate (i.e. a residue changing from the somatic allele to the germline allele), and do not acquire multiple mutations in the same step. The absence of back-mutation is justified by the relatively large number of possible mutation sites compared to the total number of mutation events.

We define the transition probability of a single mutational step from the classical fixation probability for a mutation with selection coefficient $\sigma$ in a population of size $N$(*Kimura, 1962*):

$$p_{\text{step}}(\sigma, N) = \frac{1 - e^{-\sigma}}{1 - e^{-N\sigma}}. \tag{11}$$

Here, we define the selection coefficient $\sigma$ to be proportional to the difference in log binding affinities to a particular antigen between the two sequences $s$ and $t$:

$$\sigma = \gamma \Delta_{s,t}^{\mathrm{ag}} = \gamma(-\log_{10} K_{D,t}^{\mathrm{ag}} - (-\log_{10} K_{D,s}^{\mathrm{ag}})). \tag{12}$$

This model has two tunable parameters: $N$ represents the effective population size and $\gamma$ represents how strongly differences in binding affinity impact fitness. We chose three parameter values to span a range of selection strengths (see *Figure 5—figure supplement 1*): moderate, with $N = 1000$ and $\gamma = 1$; weak, with $N = 20$ and $\gamma = 0.5$; and strong, with $N \to \infty$ and $\gamma \to \infty$ such that $p_{\mathrm{step}}$ reduces to a step function (one if $\Delta > 0$ and 0 otherwise). These three models all show similar results, with differences between selection scenarios becoming more exaggerated with stronger selection and less exaggerated with weaker selection, as expected (see *Figure 5—figure supplement 1*).

From the fixation probabilities for a given parameter regime, we have the transition probability (up to a constant factor) for all sequences $s, t$ over all antigens ag,

$$P_{s,t}^{\mathrm{ag}} = \begin{cases} p_{\mathrm{step}}(\Delta_{s,t}^{\mathrm{ag}}, \gamma, N) & \text{if } t \text{ has one more somatic mutation than } s \\ 0 & \text{otherwise} \end{cases}, \tag{13}$$

which we use for all the calculations described below for results presented in *Figure 5* and supplements.

## Scenario, mutation, and variant probabilities

It is particularly useful to store the probabilities $P_{s,t}^{\mathrm{ag}}$ as (sparse) transition matrices $P^{\mathrm{ag}}$ of dimension $2^N \times 2^N$ for each antigen, where entries are nonzero only where sequence $t$ has one more somatic mutation than $s$.

First, we wish to obtain a measure of total probability for a particular antigen scenario, as shown in *Figure 5E,F*. We calculate this by computing the matrix product over all mutational steps $i$ for a particular sequence of antigen contexts $\{\mathrm{ag}_1, \ldots, \mathrm{ag}_L\}$:

$$\mathcal{P}_{tot} = \sum_{\mathrm{paths}} \left( \prod_{\mathrm{steps}} P_{\mathrm{step}} \right) = \left[ \prod_{i=1}^{L} P^{\mathrm{ag}_i} \right]_{s_{\mathrm{g}}, s_{\mathrm{s}}}, \tag{14}$$

where $[\cdot]_{s,s'}$ corresponds to taking the matrix element in the row corresponding to variant $s$ and column corresponding to variant $s'$. In the right-most term, the products are matrix operations and $s_{\mathrm{g}}$, $s_{\mathrm{s}}$ are respectively the indices of the germline and somatic variants.

We note that the transition probabilities $P^{\mathrm{ag}_i}$ are not normalized at each step. In practice, this means that mutations are optional: many outcomes will not reach the somatic sequence and the likelihood encodes the probability of reaching the somatic state. This makes it possible to compare different scenarios, as some scenarios are more likely than others to reach the somatic state. However, because these values do not represent true probabilities — the units are arbitrary — they cannot be compared between antibodies or between selection models. The exception is for the strong scenario, where the total probability for each path is one if all steps are uphill ($\Delta_{s,t}^{\mathrm{ag}} > 0$) and 0 otherwise. Thus, here $\mathcal{P}_{tot}$ has a natural interpretation as the total number of uphill paths. When we present results from the strong model (*Figure 5C,D*, *Figure 5—figure supplement 1*, and numbers of uphill paths for H1-only scenarios as discussed in the text), we represent uphill path numbers on a linear scale without log-transforming.

Although there are many possible antigen exposure scenarios, we restrict our analysis to several classes. First, in single-antigen scenarios, all steps $i$ use the same antigen. Second, for sequential scenarios, antigen exposures must occur in non-repeating segments (for example, H1 - H3 - H1 is not allowed), although we consider all possible lengths and orders of segments.

Mixed scenarios are more complicated, as we do not fully understand the nature of B cell interactions with multiple antigens in the same germinal center (*Wang et al., 2015*; *Wang, 2017*; *Kuraoka et al., 2016*). One option is to assume that the B cell engages the antigen for which it has the highest affinity and define $\Delta$ by the maximum binding affinity across all possible antigens at each step, but this definition would trivially imply that the mixed scenario has the highest probability. Instead, we choose two alternatives: first, 'average' mixed, where we assume the B cell engages all

antigens and use the average binding affinity change over all three (for CR9114) or two (for CR6261) antigens, $\Delta_{\mathrm{mixed}} = \frac{1}{N_{\mathrm{ag}}} \sum_{\mathrm{ag}} \Delta_{\mathrm{ag}}$; and second, 'random' mixed, where we assume the B cell randomly engages a single antigen and hence the antigen at each mutational step is chosen randomly. For the latter definition, we calculate $\mathcal{P}_{tot}$ as described above for 1000 randomly drawn scenarios and average the resulting log probability. When we illustrate mutational paths and mutation orders, we choose a representative scenario (with close to median probability) from the 1000 random draws.

We estimate the error of these probabilities by bootstrapping. Specifically, for 10 bootstrap iterations, we resample each binding affinity $-\log_{10} K_{D,s}^{\mathrm{ag}}$ from a normal distribution according to its value and standard deviation. We then recalculate the total probability $\mathcal{P}_{tot}$, average over the 10 values to obtain mean and s.e.m. values, and transform by the natural log for plotting, as shown in *Figure 5* and *Figure 5—figure supplement 1*. We note that for the strong selection scenario (where probabilities represent total numbers of uphill paths), values are not log-transformed, and many scenarios have total path numbers of exactly zero. We refrain from studying the 'average' mixed scenario for strong selection because it is essentially equivalent to choosing the antigen with maximum improvement: the quantitative effect of averaging is undone when the transition probability is binarized. For CR6261, all mutations at the first mutational step are neutral (with the exception of one mutation that improves affinity for H1 only), and so we allow all mutations with equal probability for the first step in the strong selection model.

Next, to identify the most likely paths under a given exposure scenario, we reframe this Markov process as a directed weighted graph. Each sequence $s$ is a node, and a directed edge exists toward all sequences $t$ that can be reached by one additional somatic mutation. The edge weight is calculated from the transition probability, $w_{s \to t} = -\log(P_{s,t}^{\mathrm{ag}} + \epsilon)$, where $\epsilon$ is an extremely small value to ensure weights are finite. In this graph framework, we can use fast algorithms to obtain the 'shortest' paths from the germline to the somatic node (those for which the sum of weights is lowest, that is the total probability is highest). Specifically, we use the *shortest_simple_paths* function from the Python package *networkx* (*Hagberg et al., 2008*) to compute the $k$ shortest paths, as shown in *Figure 5G,H*. This method is exact and uses the algorithm described in *Yen, 1971*.

Next, we wish to obtain the probability that a mutation at site $m$ happened at a specific step $j$ (*Figure 5I,J*). As we are focusing on one antigen context, we can normalize the transition matrices and define:

$$\tilde{P}_{s,t}^{\mathrm{ag}} = P_{s,t}^{\mathrm{ag}} \times \left( \sum_t P_{s,t}^{\mathrm{ag}} \right)^{-1}, \tag{15}$$

if $P_{s,t}^{\mathrm{ag}} \neq 0$ and 0 otherwise. We can further restrict the transition matrix at step $j$, $\tilde{P}^{\mathrm{ag}_j}$, to have nonzero the mutation that occurs is at a particular residue $\alpha$, $\tilde{P}_\alpha^{\mathrm{ag}_j}$. The total relative probability for that site at that mutational step under an antigen exposure scenario is then

$$\mathcal{P}_{j,\alpha} = \left[ \left( \prod_{i=1}^{j-1} \tilde{P}^{\mathrm{ag}_i} \right) \cdot \tilde{P}_\alpha^{\mathrm{ag}_j} \cdot \left( \prod_{i=j+1}^{L} \tilde{P}^{\mathrm{ag}_i} \right) \right]_{s_g, s_s}, \tag{16}$$

where, again, products are matrix operations. Because a sequence of $L$ steps starting from the germline can only lead to the somatic state, $\tilde{P}$ verifies $\left[ \prod_{i=1}^{L} \tilde{P}^{\mathrm{ag}_i} \right]_{s_g, s_s} = 1$. With the relation $\sum_\alpha \tilde{P}_\alpha^{\mathrm{ag}_j} = \tilde{P}^{\mathrm{ag}_j}$ this implies that these probabilities are already normalized: $\sum_\alpha \mathcal{P}_{j,\alpha} = 1$.

Finally, we wish to determine the total probability of each variant (*Figure 5—figure supplement 2*), that is the sum of probabilities of all paths passing through that variant, for a given selection scenario. For a variant $s$ that contains $j$ somatic mutations, we calculate

$$\mathcal{P}_s = \left( \left[ \prod_{i=1}^{j} \tilde{P}^{\mathrm{ag}_i} \right]_{s_g, s} \right) \cdot \left( \left[ \prod_{i=j+1}^{L} \tilde{P}^{\mathrm{ag}_i} \right]_{s, s_s} \right), \tag{17}$$

where the first term is the probability of reaching sequence $s$ at mutational step $j$, and the second term is the probability of reaching the somatic sequence after passing through sequence $s$. When representing this number we add an additional normalisation factor, $\mathcal{P}'_s = \mathcal{P}_s \times n_j$, where $n_j = Lj$ is the

number of sequences with $j$ mutations, so that variants with different numbers of mutations have comparable values. $\mathcal{P}'_s$ thus represents the ratio of the probability in a selective model to the probability in a neutral model (which is $1/n_j$). Thus, sequences with $\log_{10}(\mathcal{P}'_s)>0$ are favored by the given selection scenario, and those with $\log_{10}(\mathcal{P}'_s)<0$ are disfavored, as shown in *Figure 5—figure supplement 2* for moderate selection under the optimal sequential scenario.

## Acknowledgements

We thank Rhys Adams for helpful discussion of the Tite-Seq experiments, Zach Niziolek for assistance with flow cytometry, Kevin McCarthy for help with antigen production, Matt Melissa for help acquiring strains and protocols, and Tyler Starr and members of the Denic, Gaudet, and Wittrup labs for help with experimental protocols. We also thank Jesse Bloom, Andrew Murray, and Michael Laub for helpful discussion and members of the Desai lab for comments on the manuscript. The computations in this paper were run on the FASRC Cannon cluster supported by the FAS Division of Science Research Computing Group at Harvard University.

## Additional information

### Competing interests

Aleksandra M Walczak: Senior editor, *eLife*. The other authors declare that no competing interests exist.

### Funding

| Funder | Grant reference number | Author |
| --- | --- | --- |
| Howard Hughes Medical Institute | Hanna H. Gray Postdoctoral Fellowship | Angela M Phillips |
| Hertz Foundation | Graduate Fellowship Award | Katherine R Lawrence |
| National Science Foundation | Graduate Research Fellowship Program | Katherine R Lawrence Jeffrey Chang Milo S Johnson |
| European Research Council | COG 724208 | Thierry Mora Aleksandra M Walczak |
| National Institutes of Health | GM104239 | Michael M Desai |
| National Science Foundation | PHY-1914916 | Michael M Desai |
| Stanford University | Stanford Science Fellowship | Ivana Cvijovic |
| Human Frontier Science Program | | Thomas Dupic |
| NSF-Simons Center for Mathematical and Statistical Analysis of Biology at Harvard | 1764269 | Katherine R Lawrence |

The funders had no role in study design, data collection and interpretation, or the decision to submit the work for publication.

### Author contributions

Angela M Phillips, Katherine R Lawrence, Conceptualization, Software, Formal analysis, Supervision, Validation, Investigation, Methodology, Writing - original draft, Writing - review and editing; Alief Moulana, Software, Formal analysis, Validation, Investigation, Writing - review and editing; Thomas Dupic, Software, Formal analysis, Methodology, Writing - review and editing; Jeffrey Chang, Formal analysis, Validation, Investigation, Writing - review and editing; Milo S Johnson, Software, Visualization, Writing - review and editing; Ivana Cvijovic, Conceptualization, Investigation, Methodology, Writing - review and editing; Thierry Mora, Aleksandra M Walczak, Michael M Desai, Conceptualization, Supervision, Funding acquisition, Writing - review and editing

## Author ORCIDs

Angela M Phillips https://orcid.org/0000-0002-9806-7574
Milo S Johnson https://orcid.org/0000-0003-0169-2494
Ivana Cvijovic http://orcid.org/0000-0002-6272-2979
Thierry Mora http://orcid.org/0000-0002-5456-9361
Aleksandra M Walczak http://orcid.org/0000-0002-2686-5702
Michael M Desai https://orcid.org/0000-0002-9581-1150

## Decision letter and Author response

Decision letter https://doi.org/10.7554/eLife.71393.sa1
Author response https://doi.org/10.7554/eLife.71393.sa2

# Additional files

## Supplementary files

- Supplementary file 1. Primer sequences for sequencing library preparation.
- Supplementary file 2. Fragment sequences for Golden Gate construction of the CR9114 library.
- Supplementary file 3. Primer sequences for Golden Gate construction of the CR6261 library.
- Supplementary file 4. Plasmid map of pCT302 with CR9114 germline sequence.
- Supplementary file 5. Plasmid map of pCT302 with CR9114 somatic sequence.
- Supplementary file 6. Plasmid map of pCT302 with CR6261 germline sequence.
- Supplementary file 7. Plasmid map of pCT302 with CR6261 somatic sequence.
- Supplementary file 8. Plasmid map of pFastBac with influenza A/New Caldeonia/1999 H1 ectodomain.
- Supplementary file 9. Plasmid map of pFastBac with influenza A/Hong Kong/1999 H9 ectodomain.
- Supplementary file 10. Plasmid map of pFastBac with influenza A/Wisconsin/2005 H3 ectodomain.
- Supplementary file 11. Plasmid map of pFastBac with influenza B/Ohio/2005 HA ectodomain.
- Supplementary file 12. Inferred CR9114 VH germline nucleotide sequence.
- Transparent reporting form

## Data availability

Data and code used for this study are available at https://github.com/klawrence26/bnab-landscapes (copy archived at https://archive.softwareheritage.org/swh:1:rev:61c1673a101ea739d5-b7e9b282f6bcfad41d7e90). CR9114 data are also available in an interactive data browser at https://yodabrowser.netlify.app/yoda_browser/. FASTQ files from high-throughput sequencing have been deposited in the NCBI BioProject database with accession number PRJNA741613.

The following datasets were generated:

| Author(s) | Year | Dataset title | Dataset URL | Database and Identifier |
|---|---|---|---|---|
| Phillips AM, Lawrence KR, Moulana A, Dupic T, Chang J, Johnson MS, Cvijovic I, Mora T, Walczak AM, Desai MM | 2021 | Binding affinity landscapes constrain the evolution of broadly neutralizing anti-influenza antibodies | https://www.ncbi.nlm.nih.gov/bioproject/?term=PRJNA741613 | NCBI BioProject, PRJNA741613 |
| Phillips AM, Lawrence KR, Moulana A, Dupic T, Chang J, Johnson MS, Cvijovic I, Mora T, Walczak AM, | 2021 | CR9114 Tite-Seq KD Measurements | https://yodabrowser.netlify.app/yoda_browser/ | YODA online data browser, CR9114 |

Desai MM

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

## Appendix 1

### Maximum likelihood approach to binding affinity inference

In this approach we make the assumption that the fluorescence emitted by cells of a specific genotype is distributed log-normally, with parameters $\mu_{s,c}$ and $\sigma_{s,c}$ (the mean and standard deviation of the associated normal distribution respectively). At concentration $c$, a cell with genotype $s$ will fall into the bin $b$ (log$_{10}$-fluorescence values $f_{s,c}$ ranging from $l_b$ to $h_b$) with probability:

$$P\left[l_b < f_{s,c} < h_b\right] = \int_{l_b}^{h_b} \frac{1}{\sqrt{2\pi\sigma_{s,c}^2}} e^{-\frac{1}{2}\left(\frac{f_{s,c} - \mu_{s,c}}{\sigma_{s,c}}\right)^2} \, df_{s,c} \tag{18}$$

$$= \frac{1}{2}\left(\text{erf}\left(\frac{h_b - \mu_{s,c}}{\sigma_{s,c}\sqrt{2}}\right) - \text{erf}\left(\frac{l_b - \mu_{s,c}}{\sigma_{s,c}\sqrt{2}}\right)\right). \tag{19}$$

Each cell sorted is an independent event, so the number of cells in each bin will be multinomially distributed, and thus the likelihood of sorting $n_{b,s|c}$ cells of sequence $s$ into bin $b$ at concentration $c$ is given by

$$\mathcal{L} = \prod_{s,c}\left(P\left[l_b < f_{s,c} < h_b\right]\right)^{n_{b,s|c}}, \tag{20}$$

and the log-likelihood is

$$\log\mathcal{L} = \sum_{s,c,b} n_{b,s|c} \log P\left[l_b < f_{s,c} < h_b\right] \propto \sum_{s,c,b} p_{b,s|c} \log P\left[l_b < f_{s,c} < h_b\right]. \tag{21}$$

The probability $p_{b,s|c}$ is estimated as in the mean-bin approach (see Methods) and the log-likelihood is then maximized as a function of $\mu_{s,c}$ and $\sigma_{s,c}$ (BFGS method). The values of $A$, $K_D$, and $B$ are then estimated similarly as the mean-bin approach (see Methods), replacing $\overline{F}_{s,c}$ by $\mu_{s,c}$.

The $-\log_{10}K_D$ inferred by this maximum likelihood (ML) approach correlate well with isogenic flow cytometry $-\log_{10}K_D$ (see **Appendix 1—figure 1**), but not as well as those inferred by the mean-bin approach (**Figure 1—figure supplement 2B**). The ML approach is predicated on the assumption that the fluorescence distribution for each variant is log-normal, which is often not the case (see **Appendix 1—figure 2**). For these reasons, in addition to favoring a simple approach, we performed all analyses with $-\log_{10}K_D$ inferred by the mean-bin approach.

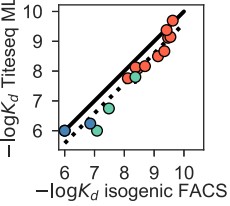

**Appendix 1—figure 1.** Correlation between $-\log_{10}K_D$ from ML inference on Tite-Seq data vs. $-\log_{10}K_D$ from isogenic flow cytometry. $-\log_{10}K_D$ to H1 (salmon), H3 (green), and Flu B (blue) shown for select variants, identical to those shown in **Figure 1—figure supplement 2B**. Pearson's r = 0.97.

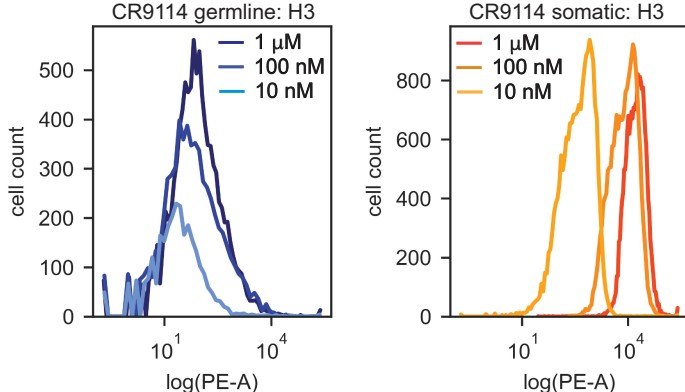

**Appendix 1—figure 2.** Distributions of PE-A fluorescence (HA binding) for isogenic CR9114 strains incubated with H3. PE-A fluorescence distributions from flow cytometry of isogenic CR9114 germline (left) and somatic (right) strains following incubation with 1 μM, 100 nM, and 10 nM H3, as described in Methods. Shape of distribution varies for different clones and is not strictly log-normal, hence deviating from assumptions made in the maximum-likelihood binding affinity inference.

## Appendix 2

### Alternative approaches to epistasis inference

Statistical epistasis and variance partitioning

The contrast between biochemical and statistical frameworks for epistasis is well described in *Poelwijk et al., 2016*. In particular, a biochemical epistasis approach highlights one particular sequence as the ''wildtype'' or reference sequence and measures effects relative to its phenotype, whereas a statistical epistasis approach measures effects relative to the average phenotype of all variants included. The biochemical approach benefits from easier interpretation of the coefficient values, particularly when there is a natural or relevant choice of reference sequence, but the coefficients at different orders are not statistically independent. The statistical approach allows for correct variance partitioning between interaction orders, but the interpretation of the coefficients can be sensitive to the set of sequences, particularly when not all possible sequences are represented or when a majority of sequences exhibit some uninteresting phenotype (e.g. lethal).

Here, we perform inference of statistical epistasis exactly as described above for biochemical epistasis (see Methods), except that genotypes $x_{i,s}$ are coded as $\{-1, 1\}$ instead of $\{0, 1\}$. The results from this statistical epistasis inference are shown in *Appendix 2—figure 1* for CR9114 and *Appendix 2—figure 2* for CR6261, in plots analogous to those in *Figure 3*, *Figure 4* and supplements. We find that the patterns of site participation in interactions are similar (although the coefficient magnitudes and signs are of course scaled differently). The group of five key sites discussed in *Figure 3* (sites 30, 57, 65, 82, and 83 for CR9114 binding to H1) exhibit coefficients that are significant for all 31 mutation combinations, consistent with the result from biochemical epistasis. Overall, the numbers of significant coefficients inferred in statistical epistasis models tends to be somewhat higher than for biochemical epistasis models, perhaps due to the effect of background averaging in reducing coefficient standard errors, but neither framework is a substantially more compact representation of epistasis than the other.

In the statistical epistasis framework, we can also partition the variance explained by the model according to the interaction order. Here, we take the final inferred model at the optimal interaction order and evaluate the prediction performance ($R^2$) of each order as a fraction of the total performance of the full model. As shown in *Appendix 2—figure 3*, we find that epistasis explains a substantial fraction of variance (18%–33%, depending on antibody-antigen pair). Variance explained tends to decline with increasing order, as is also observed in some other protein epistasis datasets (*Sailer and Harms, 2017b*). This indicates that interactions at higher order are more rare (compared to the total number of terms at each order, which scales combinatorially) and/or smaller in magnitude than those at lower order. However, this does not imply that rare, strong interactions of even higher order do not exist; for example, there may be some strong sixth-order interaction terms for CR9114 binding to H1, but not enough to compensate for the many nonsignificant sixth-order terms in our cross-validation framework.

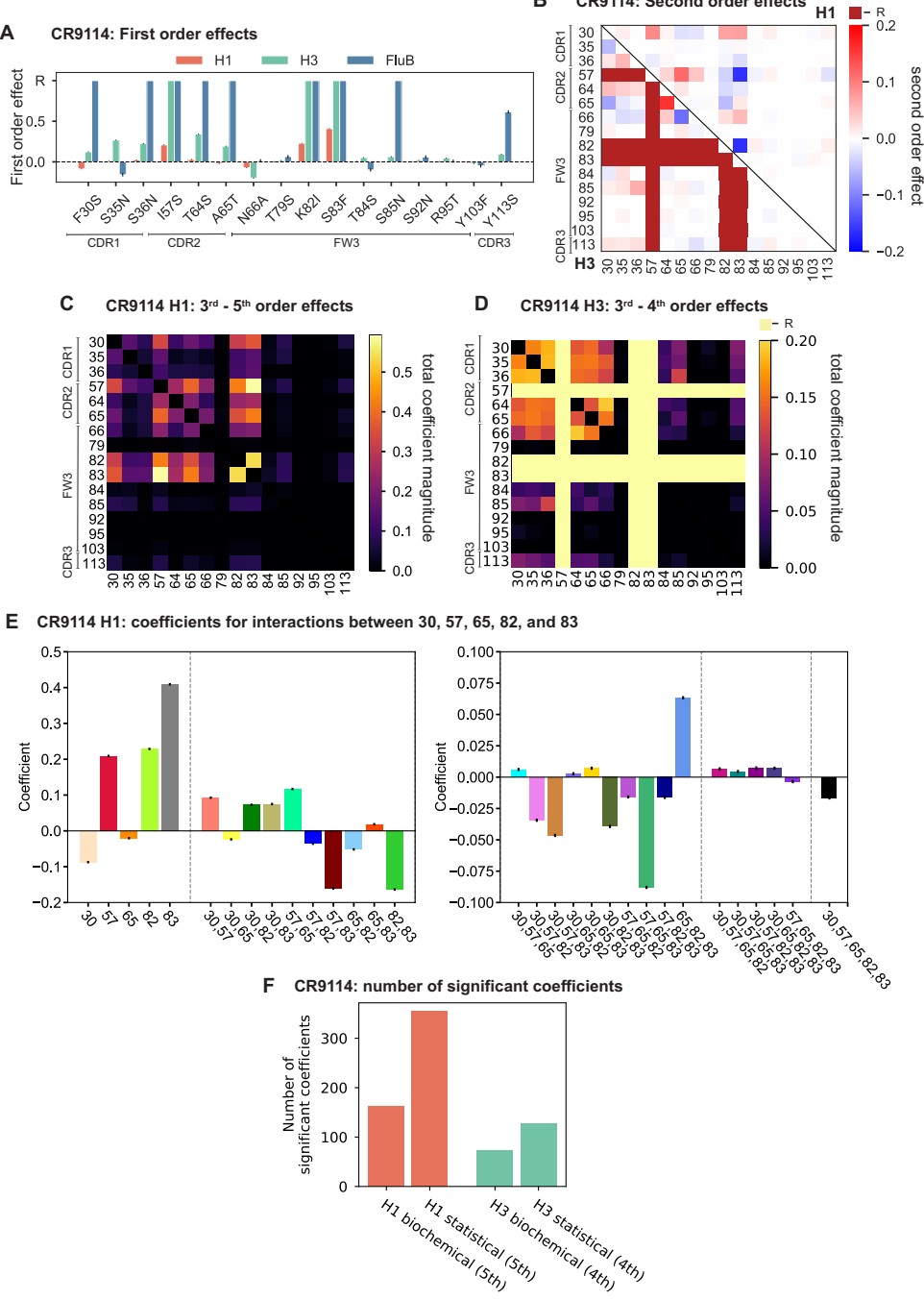

**Appendix 2—figure 1.** Results from statistical epistasis models for CR9114. (**A**) First-order effects, as in *Figure 2A*. 'R' indicates required mutations. (**B**) Second-order effects for H1 (top right) and H3 (lower left), as in *Figure 2D*. Interactions with required mutations for H3 are noted in dark red. (**C**) Cumulative higher-order effects for CR9114 binding to H1, as in *Figure 3A*. (**D**) Cumulative higher-order effects for CR9114 binding to H3, as in *Figure 3—figure supplement 3*. (**E**) Inferred interaction coefficients for the set of five key epistatic loci, as in *Figure 3—figure supplement 1B* with corresponding colors. Note the different y-axis scales for the two subplots. Different interaction orders are separated by dotted lines. (**F**) Number of significant coefficients at all orders for the biochemical and statistical epistasis models. The maximal order of interaction for each model is indicated in parentheses.

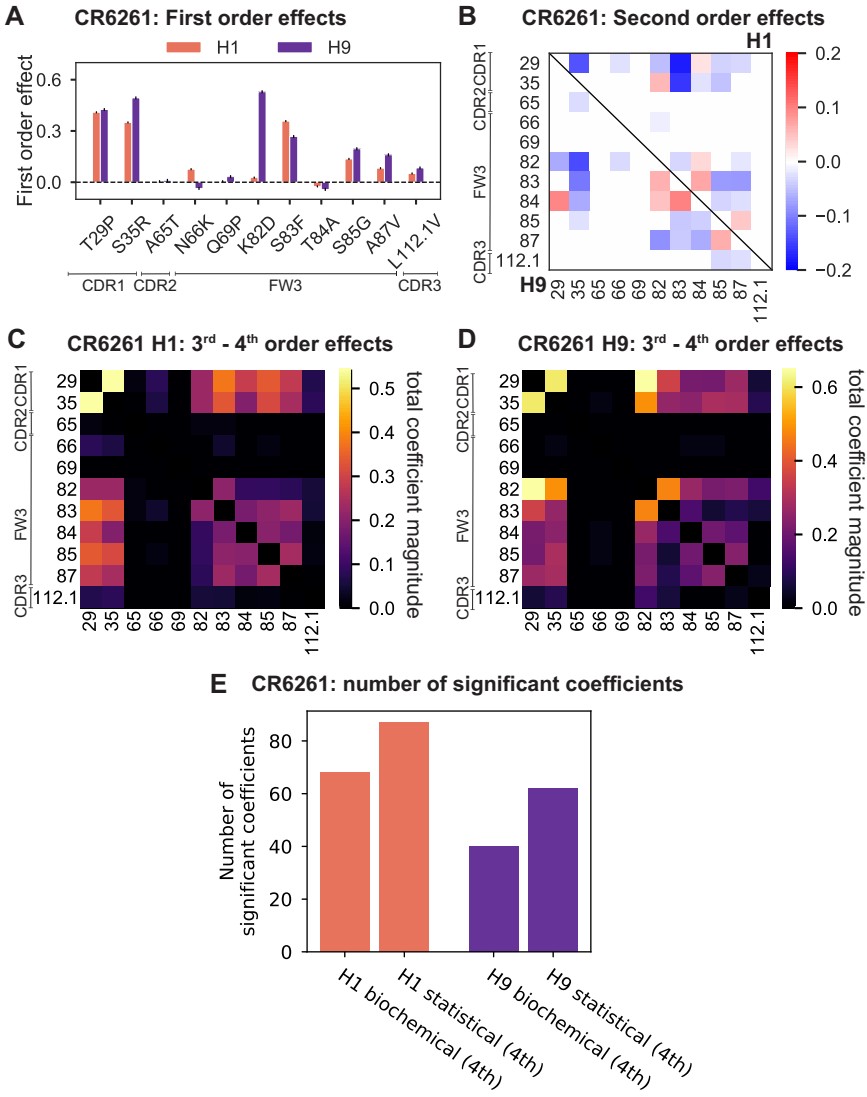

**Appendix 2—figure 2.** Results from statistical epistasis models for CR6261. (**A**) First-order effects, as in *Figure 2B*. (**B**) Second-order effects for H1 (top right) and H9 (lower left), as in *Figure 2E*. (**C**) Cumulative higher-order effects for CR6261 binding to H1, as in *Figure 4A*. (**D**) Cumulative higher-order effects for CR9114 binding to H9, as in *Figure 4—figure supplement 2A*. (**E**) Number of significant coefficients at all orders for the biochemical and statistical epistasis models. The maximal order of interaction for each model is indicated in parentheses.

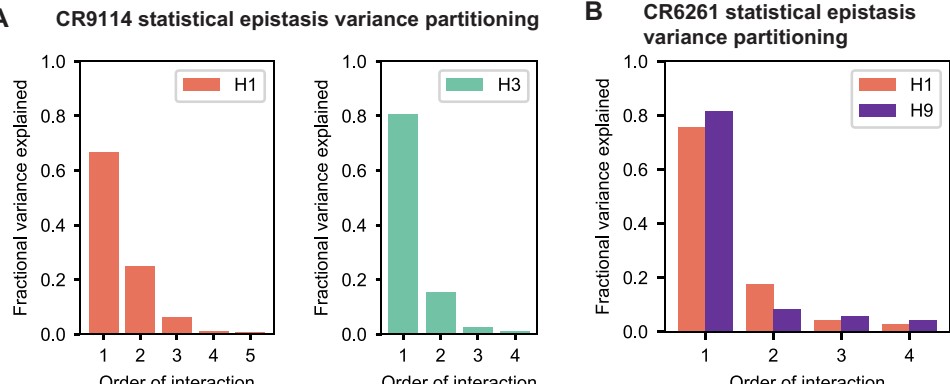

**Appendix 2—figure 3.** Variance partitioning of statistical epistasis models. (**A**) Variance partitioning for CR9114 binding to H1 (left) and H3 (right). (**B**) Variance partitioning for CR9114 binding to H1 and H9, denoted by colors as indicated.

In particular, another alternative approach to the inference of epistasis is to infer a full $L^{\text{th}}$-order model rather than truncating to lower order. This approach calculates $2^L$ epistatic coefficients, one for every datapoint, which allows for the detection of strong interactions at any order with the caveat that many coefficients may simply reflect experimental noise, especially for higher-order terms. We explore this approach by following *Poelwijk et al., 2019*: we calculate epistatic coefficients using a Walsh-Hadamard transform of the $-\log_{10} K_D$ values, and calculate standard errors on each coefficient via error propagation using the standard errors of the data. We define significant coefficients by a *p*-value cutoff of 0.05, with Bonferroni correction by the total number of parameters in the model (here $2^L$). We find that for all antibody-antigen combinations, this approach finds more significant coefficients than the optimal truncated models, many of which are at higher interaction orders than allowed in the truncated model (*Appendix 2—figure 4*). This analysis requires a measurement of $-\log_{10} K_D$ for every single variant, so we use data that has not been filtered for goodness-of-fit or error in the inference of binding affinity (see Methods), including some sequences that have substantial error. Therefore we prefer to use the more conservative regression approach for our in-depth analysis of epistasis; this inference at full order confirms the existence, strength, and identity of the high-order interactions we discuss from the regression approach, while also indicating that additional and even higher-order terms may yet exist.

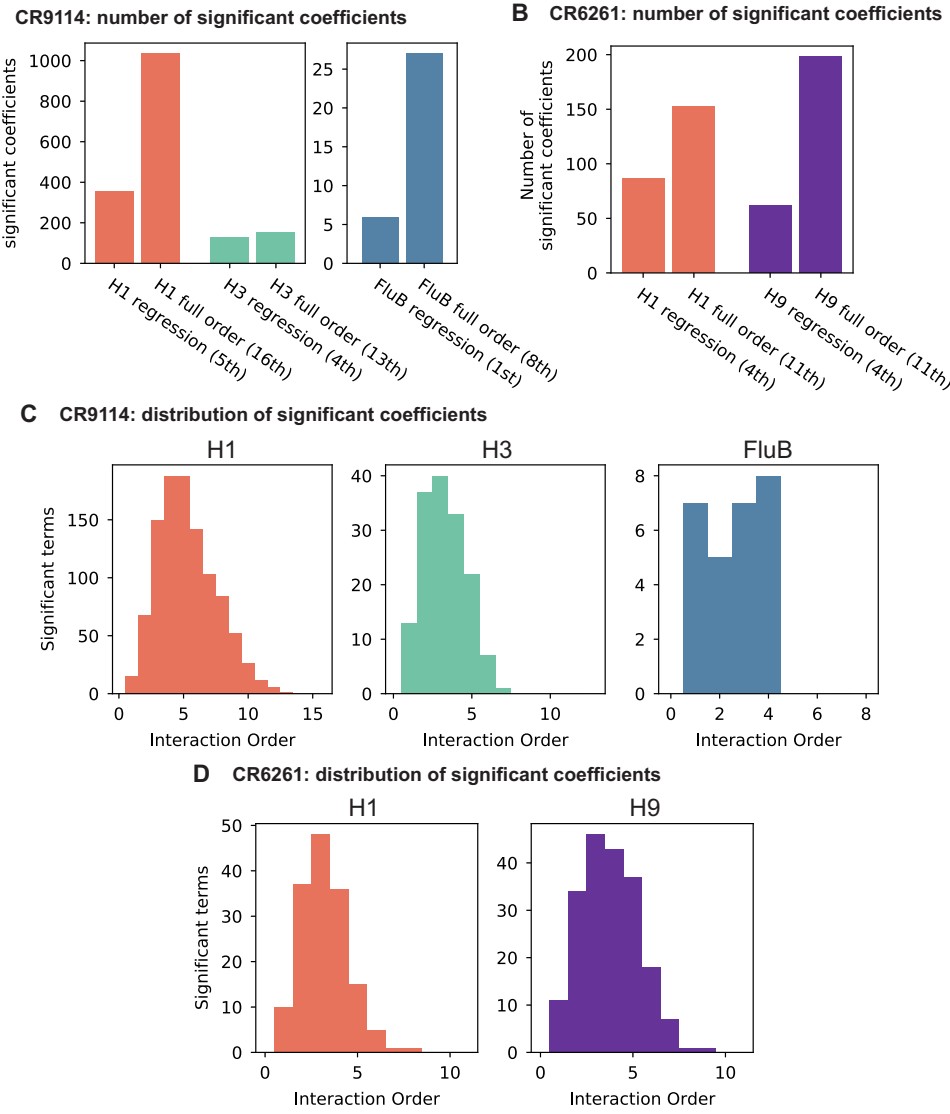

**Appendix 2—figure 4.** Epistasis inference at full order. (**A,B**) Numbers of significant coefficients for the full-order inference compared to optimal truncated regression models for (**A**) CR9114 and (**B**) CR6261. Significance for both model types is determined by $p<0.05$ with Bonferroni correction by the number of model parameters. (**C,D**) Distribution of interaction orders of significant coefficients for (**C**) CR9114 and (**D**) CR6261.

## Nonlinear models

An alternative approach to understanding epistasis is to view nonlinearities in observed phenotype data as arising from a simple nonlinear transformation applied to an underlying, unobserved additive phenotype. In this view, a simple nonlinear ''global epistasis'' function with few parameters may describe the landscape as well or better than models of the sort described above, with their large number of ''idiosyncratic epistasis'' parameters. Many studies in other proteins have attempted to disentangle such global epistasis from idiosyncratic effects (*Sailer and Harms, 2017a*; *Domingo et al., 2019*; *Sarkisyan et al., 2016*; *Otwinowski et al., 2018*; *Otwinowski, 2018*; *Adams et al., 2019*).

We already implement one global nonlinear transformation, by log-transforming our binding affinity measurements so that they are proportional to free energy changes, as described above. However, it is possible that another nonlinear transformation would capture the effects of many specific interaction coefficients, if there is a single underlying additive scale. In this section, we explore

this possibility following the approach taken by *Sailer and Harms, 2017a*: we infer a nonlinear transformation that fits the phenotype data, invert it to ''linearize'' the phenotypes, re-fit interaction models on the linearized phenotypes, and then compare those model coefficients to the original coefficients to evaluate the role of the nonlinear transformation.

Our new model is

$$y_s = \Phi\left(y_{s,\mathrm{add}}; k_m\right) = \Phi\left(\beta_0 + \sum_i^L \beta_i x_{i,s}; k_m\right), \tag{22}$$

where $y_s$ are the observed phenotypes ($-\log_{10} K_D$ values), $\Phi$ is a nonlinear function with a small number of associated parameters $k_m$, and $y_{s,\mathrm{add}}$ are the underlying additive-scale phenotypes, parametrized as before by additive coefficients $\beta_i$.

To specify $\Phi$, we must choose a family of nonlinear functions. Typical choices include splines (*Otwinowski et al., 2018*) or power transforms (*Sailer and Harms, 2017b*). We found that logistic (sigmoid) functions fit our data better than power transforms or splines, and they are monotonic and invertible. Specifically, our logistic function with four parameters is

$$\Phi(y; A, B, \mu, \sigma) = \frac{A}{1 + e^{\frac{(y-\mu)}{\sigma}}} + B. \tag{23}$$

Logistic functions capture two features that we observe: first, there is a saturation effect at low values of $-\log_{10} K_D$, corresponding to nonspecific binding that our measurements are unable to distinguish (*Batista and Neuberger, 1998*); and second, for most antibody-antigen combinations we observe a saturation effect at moderately high values of $-\log_{10} K_D$. This latter effect is not due to limits on our measurement capabilities, as illustrated by higher values of $-\log_{10} K_D$ measured for the CR6261 library to H9 compared to values of $-\log_{10} K_D$ measured for the CR9114 library to H1, but instead due to widespread ''diminishing returns'' epistasis.

After specifying the functional form of $\Phi$, we must fit both the nonlinear parameters $k_m$ and underlying linear parameters $\beta_i$. In principle, one could fit all parameters jointly, using for example a maximum likelihood approach (*Otwinowski et al., 2018*). However, we take the simpler approach as implemented in the software package from *Sailer and Harms, 2017a*, which first infers the additive parameters $\beta_i$ from the observed phenotypes and then infers the nonlinear function parameters $k_m$. We show the resulting fit of $\Phi$ in *Appendix 2—figure 5a* for two representative examples, by plotting our estimate of the additive phenotypes $y_{s,\mathrm{add}}$ on the x-axis and our observed phenotypes from data on the y-axis. We found that this simple procedure identified well-fitting $\Phi$ in a single step, and successive iterations did not significantly improve the fit.

After fitting the nonlinear transformation, we apply the inverse transformation to our observed phenotypes to obtain ''linearized'' phenotypes $y_{s,\mathrm{lin}}$:

$$y_{s,\mathrm{lin}} = \Phi^{-1}(y_s, k_m). \tag{24}$$

Because the fit of $\Phi$ is not perfect, the linearized phenotypes $y_{s,\mathrm{lin}}$ are not exactly equal to the estimated additive phenotypes $y_{s,\mathrm{add}}$, although linear regression on both quantities produces extremely similar values of $\beta_i$. For values that lie above the domain of $\Phi^{-1}$, we pin them to the largest estimated additive phenotype.

Finally, we can take our linearized phenotypes $y_{s,\mathrm{lin}}$ and infer interaction model coefficients $\beta'$ of various orders, exactly as described above for the untransformed ''raw'' phenotypes:

$$y_{s,\mathrm{lin}} = \beta'_0 + \sum_i \beta'_i x_{i,s} + \sum_{i<j}^L \beta'_{ij} x_{i,s} x_{j,s} + \sum_{i<j<k}^L \beta'_{ijk} x_{i,s} x_{j,s} x_{k,s} + \ldots + \varepsilon. \tag{25}$$

We again perform this analysis in both the biochemical and statistical epistasis frameworks. If the inverse transformation has removed most or all of the nonlinearity, then the resulting optimal interaction models should be smaller (lower maximum order of interaction and/or fewer significant interaction coefficients).

Instead, we find that in all cases, the optimal order of interaction is unchanged or only decreased by one when inferring on linearized vs raw phenotypes. Specifically, the new (vs old) optimal orders

are: 4th (vs 5th) for CR9114 binding to H1, 4th (vs 4th) for CR9114 binding to H3, 3rd (vs 4th) for CR6261 binding to H1, 3rd (vs 4th) for CR6261 binding to H9 in the biochemical epistasis framework, and 4th (vs 4th) for CR6261 binding to H9 in the statistical epistasis framework. We can compare the numbers of significant coefficients in these optimal models inferred on linearized phenotypes to the models with the same maximum order inferred on raw phenotypes (*Appendix 2—figure 5d,e*), where we see that the numbers are relatively comparable.

We next examine changes in the individual coefficients between these models. In *Appendix 2— figure 5b*, we show two representative scatterplots between the raw phenotype coefficients $\beta$ and the linearized phenotype coefficients $\beta'$, where only significant coefficients are shown for clarity. While some coefficients show dramatic changes, overall the two sets of coefficients are quite well correlated. To see which sites are involved in strong changes, we can also represent coefficient changes in a heatmap format (*Appendix 2—figure 5c*). Here, diagonal cells show the change in coefficient for single sites ($\beta'_i - \beta_i$), while off-diagonal cells show the sum of coefficient changes over all pairwise and higher terms involving each pair of mutations. We observe that for some antibody-antigen pairs, such as CR9114 binding to H1, the strongest net changes are negative, though not negative enough to remove the many significant coefficients. For other antibody-antigen pairs such as CR6261 binding to H1, there are both positive and negative net changes, indicating that the non-linear transformation is changing the epistatic landscape rather than correcting for it.

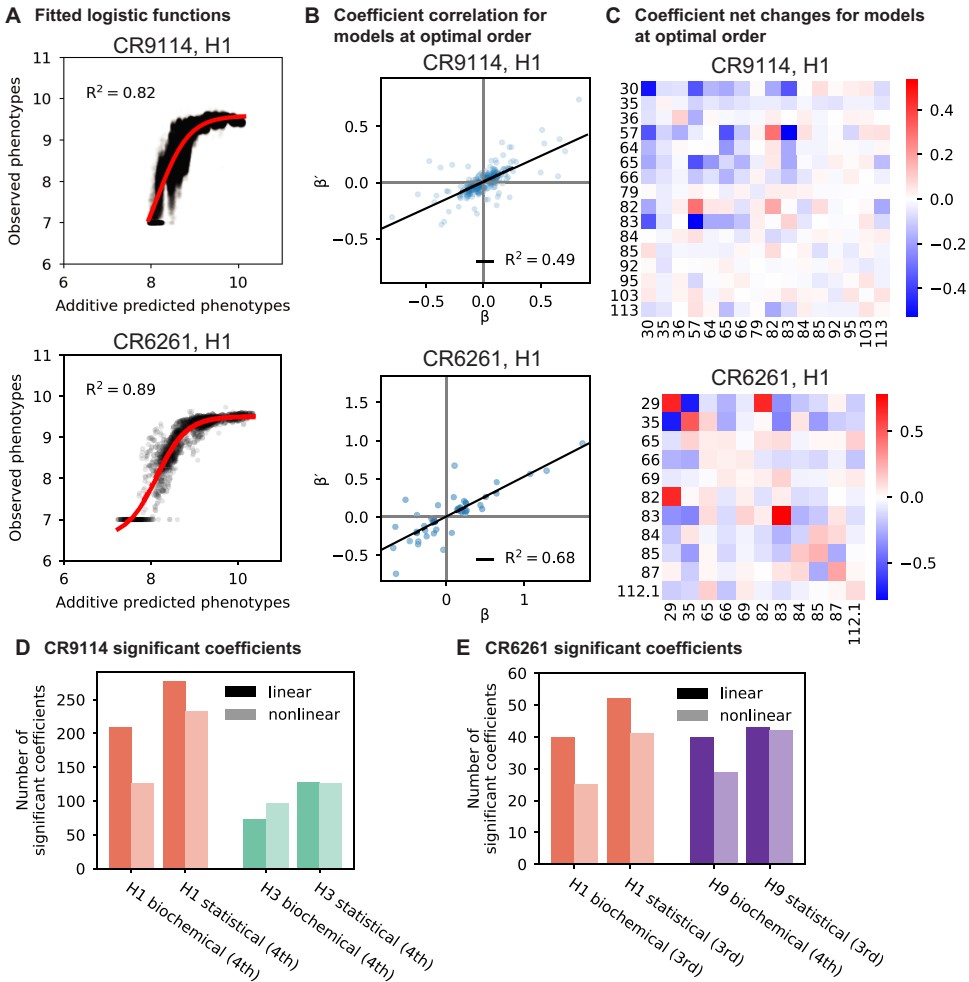

**Appendix 2—figure 5.** Results from epistasis models with nonlinear transformations. (A), Fitting logistic functions to additive predicted phenotypes. Red lines indicate the optimized logistic function Φ, with $R^2$ as indicated. (B), Scatterplot of coefficients $\beta'$ from the optimal order model inferred on linearized data (after inverting the best-fit nonlinear transformation) against original

*Appendix 2—figure 5 continued*

coefficients $\beta$ for the model with the same maximum order. (C), Net changes of coefficients by site. Diagonal cells show changes in linear coefficients. Off-diagonal cells show the sum of changes over terms at all orders (2nd and above) in which the given pair of mutations is involved. For (A-C), we show two representative antibody-antigen combinations: CR9114 binding to H1, top, and CR6261 binding to H1, bottom. (D,E), Number of significant coefficients in optimal order models fit to phenotypes transformed by the inverse nonlinear function (light bars), compared to original coefficients from linear models with the same maximal order (dark bars), for (D) CR9114 and (E) CR6261. The epistasis type and model order are indicated on the x-axis.

In summary, we find that nonlinear logistic transformations can account for a portion of the nonlinearities observed in our data, sometimes reducing the maximal order of interaction by one. However, all antigen-antibody pairs still exhibit strong idiosyncratic epistasis up to at least third order after correcting for global epistasis, and the resulting numbers and magnitudes of significant coefficients are not drastically changed. Thus, it does not appear that global epistasis can explain our data much more simply than models with individual interactions, and so we confine our main analysis to idiosyncratic epistasis models.

