## [Decision Letter]

**Acceptance summary:**

CR6261 and CR9114 are two antibodies that bind to the conserved stem of influenza hemagglutinin (HA) through their VH regions and differ by 14-18 mutations from their inferred germline sequences. The authors constructed large combinatorial libraries containing combinations of 11 and 16 mutations for CR6261 and CR9114, respectively. These were used in yeast surface display titrations to infer individual and epistatic contributions to binding diverse HAs and to infer possible evolutionary trajectories going from germline to the mature antibodies. The study provides a wealth of knowledge on amino acid contributions to binding affinity. The study informs our understanding of biochemical epistasis, and could potentially serve as a starting point for a more detailed understanding of antibody affinity maturation more generally.

**Decision letter after peer review:**

Thank you for submitting your article "Binding affinity landscapes constrain the evolution of broadly neutralizing anti-influenza antibodies" for consideration by *eLife*. Your article has been reviewed by 3 peer reviewers, and the evaluation has been overseen by a Reviewing Editor and Satyajit Rath as the Senior Editor. The following individuals involved in review of your submission have agreed to reveal their identity: Jesse D Bloom (Reviewer #2); Nicholas C Wu (Reviewer #4).

Essential revisions:

1. Following are a number of factors that may confound or limit the interpretation of the results. The authors should mention these factors and the limits they place on the analysis of evolutionary paths during antibody selection. They should also tone down the language regarding the relevance to vaccine design.

– Unlike other studies where antibodies were isolated from single-cell sorting of memory B-cells, the present bNAbs were isolated from phage display libraries. These libraries (Throsby et al., 2008; Dreyfus et al., 2012) were constructed from pooled IgM+ from 10 (not 3 as incorrectly stated in manuscript) (CR6261) or 3 (CR9114) healthy donors and scFv fragments were cloned and screened using phage display against H5HA for CR6261 (Throsby et al., 2008) and against sequential panning against Has from H1, H3 and both B lineages for CR9114 (Dreyfus et al., 2012) respectively. Use of phage display methodology which involves multiple rounds of PCR as well as panning means that mutations can be introduced during library construction. Hence the resulting sequences isolated need not accurately reflect antibody gene sequences present in the donors. Further, the multi-round panning process with diverse HAs (especially for CR9114) biases the resulting sequences selected, altering the order of the panning steps might result in a different selected sequence.

– Additionally, it is unclear what the natural selection pressures are. It is likely that they would be for increased neutralization breadth and potency which is not straightforwardly related to improved breadth of binding to soluble HA where the stem accessibility is much higher than it is in the viral context.

– The known polyreactivity of many bNAbs including stem-directed bNAbs (PMID: 33049994) is another confounding factor.

– Unlike studies with bNAbs isolated from elite neutralizers in the case of HIV-1, there is no evidence that the individuals from whom the pooled phage libaries were made had significant breadth of serum neutralizing activity and so it remains possible that the breadth of the final isolated antibodies arose from mutations introduced during the PCR amplifications and enriched through the panning steps.

2. CR6261 was selected by panning against H5 HA whereas the vaccinated individuals presumably primarily had experienced H1. Since the antibody binds to the conserved stem of HA what is relevant to interpret the mutational landscapes is not the overall antigenic diversity of HA (Figure 1B) but the diversity of the stem epitopic region. If these regions are similar in H1 and H9 HA that would explain the observation that multiple mutations in the antibody are tolerated in both cases. In the case of CR9114, the mature antibody binds best to H1, less well to H3 and weakest to B HA. Unsurprisingly the same pattern is reflected in the mutational data. Given the greater diversity in HA stem epitope sequence between H1, H3 and B relative to that between H1 and H9 it is also unsurprising that there should be the sequential acquisition of breadth in the latter CR9114 case (see also 1 above).

3. In lines 69-70, "bnAbs tend to have many more mutations than specific antibodies". This statement is mostly true for HIV bnAbs but not for influenza bnAbs. As described by Lingwood et al., (PMID: 22932267), "Influenza IGHV1-69-based broadly neutralizing antibodies undergo a relatively low degree of somatic mutation (an average of 14 amino acids in the heavy chain, n = 9)". In fact, many influenza antibodies have similar number of somatic mutations as CR9114 and CR6261 (PMID: 30795982).

4. It is surprising to see significant second-order effects at relatively large distances (Figure 2F). What is the suggested explanation?

5. It is never clearly stated in results whether it's single chain (scFv) antibody, and if the HA is trimeric. If so, is there a potential for avidity so that the measurements are Kd,apparent for the multivalent interaction rather than monomeric Kd?

Other comments:

In both the abstract and beginning of results, it would be helpful to describe the libraries a bit more clearly: all combinations of mutations separating the germline and mature antibody in the VH domain among sites contacting the epitope.

Avnir et al., PLoS Pathog 2014 (PMID: 24788925) has analyzed the somatic mutation pattern in IGHV1-69 influenza bnAbs. The authors should cite Avnir et al., to further substantiate the findings in this study. For example, Arnir et al., identified several commonly occurring somatic mutations in IGHV1-69 influenza bnAbs, including T29P, S35R, and S83F (i.e. T28P, S30R, S74F, respectively, in Kabat numbering), which have strong additive effect in CR6261 (Figure 2B).

Since evolution proceeds through single-nucleotide rather than amino acid exchanges, the germline and mature antibody nucleotide sequences should be presented (if known). The authors may consider adding a comment that evolution may not move directly through amino acid exchanges if the DNA sequence does not permit such a change through a single-nucleotide mutation.

I personally was very curious how the specific epistasis models the authors used compare to global epistasis models. This is well explained in the appendix, but might be helpful to mention in another sentence or two in main text as well.

In the non-global-epistasis model, is anything done to handle the censoring of the data at the high and low end of the affinity scale? I was confused by this because line 115 says values outside the range are pinned to the boundaries, but then in Appendix (line 1282) it seems to suggest censoring isn't an issue.

The interactive browser is cool! Would be nice if mutations were either labeled by the amino acids, or there was clearer explanation which amino acid identity is 0 and which is 1 for the violin plots. Also, might be worth adding text explaining the browser is only available for CR9114.

Line 41: I think there are actually now a lot of broadly neutralizing antibodies, so I'm not sure how accurate "a handful" is here.

Line 70: "less broad" might be better than "specific" because even bNAbs are specific (for flu HA for instance) rather than "sticky" to everything.

I had a hard time understanding the numbers in Figure 1A.

I found Figure 1G confusing, mostly because I didn't (and still don't) really understand all the boxes with solid and dotted black lines labeled with various HAs.

Line 288-289: I think we can be virtually certain that the person wasn't infected with H9 since it's not a human virus subtype.

It would be useful to list the surface area each residue in the paratope for both antibodies contributes to binding where such information is available.

Is it possible to convert the effect scores into a free energy of binding contribution? Are there error estimates for the effect scores that would allow one to assess whether apparent differences in effect scores are statistically significant?

Larger epistatic effect scores occur between residues with the largest contributions to binding. Is this expected?

The authors may consider changing "CR-9114" and "CR-6261" to "CR9114" and "CR6261", respectively, which are the more commonly used nomenclatures for these two antibodies in the field.

It is unclear to me which antibody numbering scheme (e.g. Kabat, Chothia, IMGT) is being used for numbering amino acid residues in this study. Also, naming a position as "112.1" in CR6261 seems a bit odd. Should that be "112a"?

In panel B of Figure 1—figure supplement 5, the correlation between Tite-Seq mean expression and isogenic expression fluorescence is quite high for CR9114 variants but low for CR6261 variants. Is there a reason for that discrepancy?

In lines 224-225, "In contrast, a specific set of many mutations with strong synergistic interactions is required to bind H3, and to an even greater extent, influenza B". It is not very clear to me which analysis supports this claim. Can the authors clarify a bit?

In caption of Figure 5, while the meaning of "R" is described (random mixed scenario), the meaning of "O" and "A" should also be explained.

Among the somatic mutations in CR9114, I57S, K82I, and S83F are particularly important for binding to H3 (lines 182-183 and Figure 2D). Can the author provide a plausible structural explanation?

Similar to the comment above, it is interesting to see that T29P almost exclusively occurs as the first mutation under various selection scenarios (Figure 5J and Figure 5—figure supplement 3C-D). Can the author provide a plausible structural explanation?

*Reviewer #1 Recommendations for the authors:*

The interactive browser is cool! It would be nice if mutations were either labeled by the amino acids, or if there was a clearer explanation which amino acid identity is 0 and which is 1 for the violin plots. Also, might be worth adding text explaining the browser is only available for CR9114.

Line 41: I think there are actually now a lot of broadly neutralizing antibodies, so I'm not sure how accurate "a handful" is here.

Line 70: "less broad" might be better than "specific" because even bNAbs are specific (for flu HA for instance) rather than "sticky" to everything.

I had a hard time understanding the numbers in Figure 1A.

I found Figure 1G confusing, mostly because I didn't (and still don't) really understand all the boxes with solid and dotted black lines labeled with various HAs.

Line 288-289: I think we can be virtually certain that the person wasn't infected with H9 since it's not a human virus subtype.

*Reviewer #4 Recommendations for the authors:*

1. The authors may consider changing "CR-9114" and "CR-6261" to "CR9114" and "CR6261", respectively, which are the more commonly used nomenclatures for these two antibodies in the field.

2. It is unclear to me which antibody numbering scheme (e.g. Kabat, Chothia, IMGT) is being used for numbering amino acid residues in this study. Also, naming a position as "112.1" in CR6261 seems a bit odd. Should that be "112a"?

3. In lines 69-70, "bnAbs tend to have many more mutations than specific antibodies". This statement is mostly true for HIV bnAbs but not for influenza bnAbs. As described by Lingwood et al., (PMID: 22932267), "Influenza IGHV1-69-based broadly neutralizing antibodies undergo a relatively low degree of somatic mutation (an average of 14 amino acids in the heavy chain, n = 9)". In fact, many influenza antibodies have similar number of somatic mutations as CR9114 and CR6261 (PMID: 30795982).

4. In panel B of Figure 1—figure supplement 5, the correlation between Tite-Seq mean expression and isogenic expression fluorescence is quite high for CR9114 variants but low for CR6261 variants. Is there a reason for that discrepancy?

5. In lines 224-225, "In contrast, a specific set of many mutations with strong synergistic interactions is required to bind H3, and to an even greater extent, influenza B". It is not very clear to me which analysis supports this claim. Can the authors clarify a bit?

6. In caption of Figure 5, while the meaning of "R" is described (random mixed scenario), the meaning of "O" and "A" should also be explained.

7. Among the somatic mutations in CR9114, I57S, K82I, and S83F are particularly important for binding to H3 (lines 182-183 and Figure 2D). Can the author provide a plausible structural explanation?

8. Similar to the comment above, it is interesting to see that T29P almost exclusively occurs as the first mutation under various selection scenarios (Figure 5J and Figure 5—figure supplement 3C-D). Can the author provide a plausible structural explanation?

9. Avnir et al., PLoS Pathog 2014 (PMID: 24788925) has analyzed the somatic mutation pattern in IGHV1-69 influenza bnAbs. The authors should cite Avnir et al., to further substantiate the findings in this study. For example, Arnir et al., identified several commonly occurring somatic mutations in IGHV1-69 influenza bnAbs, including T29P, S35R, and S83F (i.e. T28P, S30R, S74F, respectively, in Kabat numbering), which have strong additive effect in CR6261 (Figure 2B).

---

## [Author Response]

Essential revisions:1. Following are a number of factors that may confound or limit the interpretation of the results. The authors should mention these factors and the limits they place on the analysis of evolutionary paths during antibody selection. They should also tone down the language regarding the relevance to vaccine design.

We have toned down the language regarding the relevance to vaccine design. We have also addressed each of the specific caveats:

– Unlike other studies where antibodies were isolated from single-cell sorting of memory B-cells, the present bNAbs were isolated from phage display libraries. These libraries (Throsby et al., 2008; Dreyfus et al., 2012) were constructed from pooled IgM+ from 10 (not 3 as incorrectly stated in manuscript) (CR6261) or 3 (CR9114) healthy donors and scFv fragments were cloned and screened using phage display against H5HA for CR6261 (Throsby et al., 2008) and against sequential panning against Has from H1, H3 and both B lineages for CR9114 (Dreyfus et al., 2012) respectively. Use of phage display methodology which involves multiple rounds of PCR as well as panning means that mutations can be introduced during library construction. Hence the resulting sequences isolated need not accurately reflect antibody gene sequences present in the donors. Further, the multi-round panning process with diverse HAs (especially for CR9114) biases the resulting sequences selected, altering the order of the panning steps might result in a different selected sequence.

We have commented on the potentially confounding factors associated with phage display in the Discussion. We note that however biased the discovery of these sequences may have been, these antibodies occupy uniquely interesting regions of sequence space because of their associated breadth and are thus useful for understanding the relationship between sequence, affinity, and breadth. We have also made the noted correction to the Methods section (i.e. that CR6261 was isolated from pooled PBMC from 10 donors rather than 3), and thank the reviewer for bringing this to our attention.

– Additionally, it is unclear what the natural selection pressures are. It is likely that they would be for increased neutralization breadth and potency which is not straightforwardly related to improved breadth of binding to soluble HA where the stem accessibility is much higher than it is in the viral context.

We have commented on this potential caveat in the Discussion and have referred to several studies that find affinity to soluble recombinant HA to be indicative of viral neutralization.

– The known polyreactivity of many bNAbs including stem-directed bNAbs (PMID: 33049994) is another confounding factor.

Selection against polyreactivity is likely an important factor shaping bnAb maturation that is not measured in this study and thus is not captured by our models. We have commented on this potential caveat in the Discussion, referring to studies examining the polyreactivity of stem-targeting bnAbs.

– Unlike studies with bNAbs isolated from elite neutralizers in the case of HIV-1, there is no evidence that the individuals from whom the pooled phage libaries were made had significant breadth of serum neutralizing activity and so it remains possible that the breadth of the final isolated antibodies arose from mutations introduced during the PCR amplifications and enriched through the panning steps.

We have added a comment on the potential caveats associated with phage display PCR and panning to the Discussion.

2. CR6261 was selected by panning against H5 HA whereas the vaccinated individuals presumably primarily had experienced H1. Since the antibody binds to the conserved stem of HA what is relevant to interpret the mutational landscapes is not the overall antigenic diversity of HA (Figure 1B) but the diversity of the stem epitopic region. If these regions are similar in H1 and H9 HA that would explain the observation that multiple mutations in the antibody are tolerated in both cases. In the case of CR9114, the mature antibody binds best to H1, less well to H3 and weakest to B HA. Unsurprisingly the same pattern is reflected in the mutational data. Given the greater diversity in HA stem epitope sequence between H1, H3 and B relative to that between H1 and H9 it is also unsurprising that there should be the sequential acquisition of breadth in the latter CR9114 case (see also 1 above).

We agree that the diversity of the stem epitope is important for comparing these antigens and have included this information in Figure 1—Figure supplement 1 and have referenced this in main text. In Figure 1B we retained the antigenic diversity of HA to illustrate the varying levels of breadth of the two antibodies, to motivate the choice of HA antigen for screening each antibody library, and to orient readers unfamiliar with influenza groups and HA subtypes. While we agree that retrospectively the nested sets of mutations required for CR9114 breadth may seem unsurprising given CR9114 affinity for each of these antigens and the diversity of the corresponding stem epitopes, this finding is novel and provides new molecular insight into the constraints on mutational pathways to antibodies with varying levels of breadth.

3. In lines 69-70, "bnAbs tend to have many more mutations than specific antibodies". This statement is mostly true for HIV bnAbs but not for influenza bnAbs. As described by Lingwood et al., (PMID: 22932267), "Influenza IGHV1-69-based broadly neutralizing antibodies undergo a relatively low degree of somatic mutation (an average of 14 amino acids in the heavy chain, n = 9)". In fact, many influenza antibodies have similar number of somatic mutations as CR9114 and CR6261 (PMID: 30795982).

We have removed this statement.

4. It is surprising to see significant second-order effects at relatively large distances (Figure 2F). What is the suggested explanation?

Second-order effects could be caused by direct interactions between neighboring residues but also by indirect interactions between residues, either through their interactions with antigen or through conformational rearrangements. The second-order effects we observe at relatively large distances likely result from indirect interactions, and we have clarified this in the text.

5. It is never clearly stated in results whether it's single chain (scFv) antibody, and if the HA is trimeric. If so, is there a potential for avidity so that the measurements are Kd,apparent for the multivalent interaction rather than monomeric Kd?

We have clarified that the constructs are scFv antibodies and that the HA is trimeric in the results. In regard to potential avidity, due to the density of scFv antibodies on the yeast cell (10^4^-10^5^, Bodor and Wittrup 1997) and the surface area of a budding yeast cell (60 mm^2^, Klis et al., 2014), it is unlikely that two scFv bind a single HA trimer (distance between stem epitopes on adjacent monomers < 10 nm). This potential issue is further avoided by sorting single cells, such that scFv on different yeast cells that could potentially interact with a single HA trimer would be discarded during sorting. Thus, we expect that the K_D_ measurements made here reflect monomeric binding interactions between trimeric HA and a single scFv. Additionally, the original development of the yeast display system used here established that this method enables measurement of equilibrium dissociation constants (K_D_) that are comparable to solution-based measurements (Bodor and Wittrup 1997; Bodor and Wittrup 2003).

Other comments:In both the abstract and beginning of results, it would be helpful to describe the libraries a bit more clearly: all combinations of mutations separating the germline and mature antibody in the VH domain among sites contacting the epitope.

We have edited the abstract to specify that the mutations are located in the heavy-chain for both antibodies. In the results, we have clarified that we are examining mutations separating the germline and somatic sequences. Although we did exclude 2 (CR9114) and 3 (CR6261) distant framework mutations from our libraries, we did not limit our libraries to sites contacting the epitope.

Avnir et al., PLoS Pathog 2014 (PMID: 24788925) has analyzed the somatic mutation pattern in IGHV1-69 influenza bnAbs. The authors should cite Avnir et al., to further substantiate the findings in this study. For example, Arnir et al., identified several commonly occurring somatic mutations in IGHV1-69 influenza bnAbs, including T29P, S35R, and S83F (i.e. T28P, S30R, S74F, respectively, in Kabat numbering), which have strong additive effect in CR6261 (Figure 2B).

We thank reviewers for pointing out this study and have added this citation to our discussion of single mutational effects.

Since evolution proceeds through single-nucleotide rather than amino acid exchanges, the germline and mature antibody nucleotide sequences should be presented (if known). The authors may consider adding a comment that evolution may not move directly through amino acid exchanges if the DNA sequence does not permit such a change through a single-nucleotide mutation.

The mature nucleotide sequences are published and the associated GenBank IDs have been added to our Materials and methods section for additional clarity. The germline nucleotide sequences are not published but can be inferred from the mature sequences using tools such as IMGT or IgBlast (as we did here for CR9114; we have added a description of this process to our Materials and methods). We have included the inferred germline heavy chain nucleotide sequence for CR9114 as Supplementary file 12. For CR6261, we did not infer the germline nucleotide sequence but instead designed our libraries based on the published germline amino acid sequence (Lingwood et al., 2012). Further, all nucleotide sequences used here to generate the antibody libraries (available as plasmid maps in Supplementary files 4-7) were codon optimized for expression in yeast. Thus, the nucleotide sequences of our libraries are relevant for examining evolution at the amino acid level, not the nucleotide level. Still, based on the inferred germline sequence for CR9114, several (4/16) of the amino acid changes occurred via multiple nucleotide mutations, which may occur in a single round of somatic hypermutation (Spisak 2020, Unniraman 2007), or over two distinct rounds. A comment to reflect this point has been added to the ‘Affinity to diverse antigens was likely acquired sequentially’ section of the Results.

I personally was very curious how the specific epistasis models the authors used compare to global epistasis models. This is well explained in the appendix, but might be helpful to mention in another sentence or two in main text as well.

We have added a sentence summarizing the results from the global epistasis analysis, in the paragraph where we introduce the epistasis modeling approach.

In the non-global-epistasis model, is anything done to handle the censoring of the data at the high and low end of the affinity scale? I was confused by this because line 115 says values outside the range are pinned to the boundaries, but then in Appendix (line 1282) it seems to suggest censoring isn't an issue.

For all epistasis analysis approaches, we pin inferred –logK_D_ values outside of the titration range to our titration boundaries, as other Tite-Seq experiments have done previously (Adams et al., 2016, 2019). For the high-affinity boundary (10^-11^ or 10^-12^ M), we never observe binding affinities within an order of magnitude of the boundary, so pinning does not occur. The Appendix sentence in question is referring to the “diminishing returns” pattern among high-affinity variants, which therefore is due to epistasis rather than data censoring. However, at the low-affinity boundary, we do observe many variants with no detectable binding, especially for CR9114 binding to H3 and influenza B. As these concentrations are high enough that nonspecific binding becomes appreciable (Batista 1998), we believe it is not meaningful to assign –logK_D_ values lower than the boundary (6 or 7 depending on the antigen). Rather than discard these variants altogether, we choose to preserve some measure of their low binding affinity (albeit biased) by assigning them the boundary value and proceeding with inference as described.

The interactive browser is cool! Would be nice if mutations were either labeled by the amino acids, or there was clearer explanation which amino acid identity is 0 and which is 1 for the violin plots. Also, might be worth adding text explaining the browser is only available for CR9114.

We are clarifying the amino acid identity on the violin plots in the data browser (will be completed prior to publication). We have also referred to the data browser in the legend of Figure 3 and in the Data Availability statement we specify that the browser is only available for CR9114 (we find that the smaller size of the CR6261 dataset makes the data browser less helpful).

Line 41: I think there are actually now a lot of broadly neutralizing antibodies, so I'm not sure how accurate "a handful" is here.

This language was chosen to reflect the rare frequency of bnAbs in natural repertoires. Upon review we agree that this is confusing and have changed this to ‘numerous’.

Line 70: "less broad" might be better than "specific" because even bNAbs are specific (for flu HA for instance) rather than "sticky" to everything.

We have made this change.

I had a hard time understanding the numbers in Figure 1A.

To clarify the numbering in Figure 1A we added lines to connect the residue number to the corresponding amino acid. This numbering is IMGT unique numbering, which is defined such that different receptors and chains can be easily compared. We have added a note to the figure caption regarding the numbering system.

I found Figure 1G confusing, mostly because I didn't (and still don't) really understand all the boxes with solid and dotted black lines labeled with various HAs.

We agree that the caption does not clearly explain Figure 1G and have added more detail to explain the insets.

Line 288-289: I think we can be virtually certain that the person wasn't infected with H9 since it's not a human virus subtype.

Although we agree that infection with H9 is unlikely, we also don’t wish to imply that we can definitely rule it out. We have adjusted the language to 'this breadth was likely acquired by chance’.

It would be useful to list the surface area each residue in the paratope for both antibodies contributes to binding where such information is available.

This information is not available in the published structural characterizations of these antibodies (Ekiert et al., 2009, Dreyfus et al., 2012), but we have calculated this using ChimeraX (Pettersen 2021) and find that it correlates well with our previous analysis of the number of proximal HA residues (both metrics are now available in Figure 2–source data 3). Because it is a more standard calculation, we decided to plot this contact surface area in Figure 2C and Figure 2—figure supplement 1 instead of the number of proximal HA residues, and we thank reviewers for this suggestion.

Is it possible to convert the effect scores into a free energy of binding contribution? Are there error estimates for the effect scores that would allow one to assess whether apparent differences in effect scores are statistically significant?

The –logK_D_ values are directly proportional to DG_binding_, as DG = –RTlnK, and mutation coefficients are inferred from linear regression on these values. Thus effect scores (additive and epistatic) are directly proportional to changes in free energy of binding upon mutation. We have clarified this relationship in the main text. For all effect scores, the standard error, p-value, and 95% confidence interval resulting from ordinary least squares regression (with Bonferroni correction by the number of parameters) is provided in Figure 2–source data 1 (CR9114) and Figure 2–source data 2 (CR6261). In all figures, we either display effect score standard errors as error bars where possible (eg Figure 2A,B, Figure 3F) or we display only statistically significant effect scores (assessed as a Bonferroni-corrected p-value < 0.05, eg Figure 2D,E, Figure 3A), as indicated in the respective captions.

Larger epistatic effect scores occur between residues with the largest contributions to binding. Is this expected?

This is not necessarily expected. In our datasets, this is often true but is not always the case, as illustrated by the strong third-order interaction between CR9114 mutations 57, 64, and 65, which have only weak individual effects (Figure 3E). We also do not expect this effect to arise as an artifact of our statistical inference approach, as we jointly infer coefficients at all orders for all sites.

The authors may consider changing "CR-9114" and "CR-6261" to "CR9114" and "CR6261", respectively, which are the more commonly used nomenclatures for these two antibodies in the field.

We have made this change.

It is unclear to me which antibody numbering scheme (e.g. Kabat, Chothia, IMGT) is being used for numbering amino acid residues in this study. Also, naming a position as "112.1" in CR6261 seems a bit odd. Should that be "112a"?

This numbering is IMGT unique numbering, which is defined such that different receptors and chains can be easily compared. We have added a note to the figure caption regarding the numbering system.

In panel B of Figure 1—figure supplement 5, the correlation between Tite-Seq mean expression and isogenic expression fluorescence is quite high for CR9114 variants but low for CR6261 variants. Is there a reason for that discrepancy?

These specific variants were selected to span a wide range of –logK_D_ to validate the Tite-Seq measurements with isogenic measurements. Across both CR9114 and CR6261 libraries, the variation in expression is quite modest and thus has relatively low correlation between replicate Tite-Seq measurements (Figure 1—figure supplement 5A) and with isogenic fluorescence measurements (Figure 1—figure supplement 5B). For this reason, and because the antibody sequences are displayed on yeast in an scFv format, we do not draw major conclusions from the expression data. We suspect that the higher correlation for CR9114 variants is related to the specific variants chosen, rather than a general feature of the CR9114 library (e.g. CR9114 S83F was one of the isogenic variants and has the strongest impact on expression; CR6261 S83F also has the strongest impact on expression but was not reconstructed for isogenic measurements). Still, we present these data to demonstrate that (1) scFv expression was not strongly impacted by sequence identity and that (2) some mutations have modest effects on expression that may be inversely correlated their impact on binding affinity (Figure 1—figure supplement 5E). The second point is statistically significant when examining all sequences in a given library (Figure 1—figure supplement 5E), and although we refrain from equating yeast scFv expression with antibody stability, this finding may be of interest to others in the protein evolution community and is generally consistent with other studies of protein evolution (Tokuriki and Tawfik 2009).

In lines 224-225, "In contrast, a specific set of many mutations with strong synergistic interactions is required to bind H3, and to an even greater extent, influenza B". It is not very clear to me which analysis supports this claim. Can the authors clarify a bit?

We have added a reference to Figure 2A, which shows the nested sets of required mutations for binding H3 and influenza B, to support this claim more clearly.

In caption of Figure 5, while the meaning of "R" is described (random mixed scenario), the meaning of "O" and "A" should also be explained.

This has been added to the caption of Figure 5.

Among the somatic mutations in CR9114, I57S, K82I, and S83F are particularly important for binding to H3 (lines 182-183 and Figure 2D). Can the author provide a plausible structural explanation?

CR9114 binds H3 through the flexibility of its combining site, which enables it to make favorable interactions with different conformations of Trp21 and accommodate Asn49 as well as the Asn38 glycan in group 2 HAs (Dreyfus et al., 2012). Here, we find that K82I and S83F are important for binding H1, H3 and influenza B HA, which is likely due to hydrophobic interactions between these mutated residues and a conserved hydrophobic patch in HA. Similarly, we find that I57S is important for binding each antigen, though for H1 it only improves affinity in specific sequence backgrounds. I57S is in CDR2 of CR9114, and although the contact surface area with H3 is quite modest (Figure 2—figure supplement 1), it may enhance binding of CR9114 to H3 by reorienting neighboring residues in the CDR2 loop, which interacts directly with Trp21 and Asn49, and likely accommodates the glycan at Asn38. Because we do not have the co-crystal structure with the germline antibody, we can only speculate on the impact of each mutation and thus have added a brief plausible structural explanation to the text.

Similar to the comment above, it is interesting to see that T29P almost exclusively occurs as the first mutation under various selection scenarios (Figure 5J and Figure 5—figure supplement 3C-D). Can the author provide a plausible structural explanation?

T29P is the only single mutation that confers any detectable H1 binding affinity and is thus the most likely mutation to occur first in all selection models that begin with H1. Importantly, the affinity of this single mutant is only modestly improved relative to the germline sequence (–logK_D_ = 7.53 +/- 0.04, compared to 7.0 +/- 0.0 for the germline). Still, this improvement is significant and reflects prior structural characterization of CR6261, which demonstrates that CDR1 (where T29P is located) makes extensive hydrogen bonding interactions with the A helix of HA2 (Ekiert et al., 2009). The contact area between Pro29 and HA are annotated in Figure 2C, and in the pathways analysis section of the results, we have added a sentence to clarify the strong preference for this mutation to occur first.